# Janus nanoparticles targeting extracellular polymeric substance achieve flexible elimination of drug-resistant biofilms

Zhiwen Liu[1,2,3,4], Kangli Guo[1,2,3,4], Liemei Yan[1,2,3], Kai Zhang[1,2,3], Ying Wang[1,2,3], Xiaokang Ding[1,2,3], Nana Zhao [1,2,3] ✉ & Fu-Jian Xu [1,2,3] ✉

Safe and efficient antibacterial materials are urgently needed to combat drug-resistant bacteria and biofilm-associated infections. The rational design of nanoparticles for flexible elimination of biofilms remains challenging. Herein, we propose the fabrication of Janus-structured nanoparticles targeting extracellular polymeric substance to achieve dispersion or near-infrared (NIR) light-activated photothermal elimination of drug-resistant biofilms, respectively. Asymmetrical Janus-structured dextran-bismuth selenide (Dex-BSe) nanoparticles are fabricated to exploit synergistic effects of both components. Interestingly, Janus Dex-BSe nanoparticles realize enhanced dispersal of biofilms over time. Alternatively, taking advantage of the preferential accumulation of nanoparticles at infection sites, the self-propelled active motion induced by the unique Janus structure enhances photothermal killing effect. The flexible application of Janus Dex-BSe nanoparticles for biofilm removal or NIR-triggered eradication in vivo is demonstrated by *Staphylococcus aureus*-infected mouse excisional wound model and abscess model, respectively. The developed Janus nanoplatform holds great promise for the efficient elimination of drug-resistant biofilms in diverse antibacterial scenarios.

Due to the increase in antibiotic resistance, bacterial infections pose a serious threat to human health[1,2]. There is an urgent need for alternative antibacterial strategies to combat drug-resistant bacteria and biofilm-associated infections in diverse scenarios. Some antibacterial materials such as cationic polymers[3,4], antimicrobial peptides[5], and metal ions[6] have been used to fight against drug-resistant bacterial infections. In addition, it is noteworthy that more than 80% of bacterial infections are related to biofilms[7,8]. Compared with planktonic bacteria, the shielding matrix of extracellular polymeric substance (EPS) in biofilms can not only protect the bacterial cells from host immune defenses, but also prevent the penetration of antibacterial materials[9–11]. Near-infrared (NIR) light-responsive photothermal therapy (PTT) possesses the advantages of minimal invasiveness and not restricted by the microenvironment, which demonstrates great potential in the treatment of drug-resistant bacteria and eradication of biofilms[12–17]. A variety of nanoparticles with photothermal properties have been employed to damage the bacteria through localized hyperthermia[12,18]. However, the limited penetration of photothermal nanoparticles in the biofilm greatly impedes the therapeutic efficacy of PTT. Therefore, it remains a great challenge to develop efficient photothermal nanoparticles for the flexible treatment of both drug-resistant bacteria and biofilm-related infections to meet the needs of different antibacterial scenarios.

As a bacterial polysaccharide with high stability and biocompatibility, dextran has been widely applied in biomedical fields[19–23]. In particular, dextran could be incorporated in the matrix of biofilms by

[1]State Key Laboratory of Chemical Resource Engineering, Beijing University of Chemical Technology, Beijing 100029, China. [2]Key Laboratory of Biomedical Materials of Natural Macromolecules (Beijing University of Chemical Technology), Ministry of Education, Beijing 100029, China. [3]Beijing Laboratory of Biomedical Materials, Beijing University of Chemical Technology, Beijing 100029, China. [4]These authors contributed equally: Zhiwen Liu, Kangli Guo. ✉e-mail: zhaonn@mail.buct.edu.cn; xufj@mail.buct.edu.cn

exoenzymes for the synthesis of EPS glucans and diffuse into biofilms[19,22]. Dextran-modified nanoparticles were found to target biofilms with high specificity and enhanced penetration[19,22]. More interestingly, positively charged polymeric nanoparticles with the dextran shell can achieve biofilm dispersion by nanoscale bacterial debridement through enhanced solubility of the bacteria-nanoparticle complex[24], which is considered promising for biofilm removal in clinical applications, such as wound dressings and disinfectant rinses[25,26]. While the unique biofilm removal mechanism makes the strategy effective for the eradication of multidrug-resistant biofilm, the underlying molecular mechanism for the biofilm dispersion remains exploration. It would be desirable to construct a nanoplatform with an exposed surface of dextran, which combines excellent biofilm targeting, penetration, and dispersion for advanced antibacterial therapy with a clear molecular mechanism.

Currently, most of the photothermal nanoparticles used in biomedical applications are symmetrical spherical shapes[13,15,16,27–31]. On the other hand, asymmetric hetero-nanostructures have attracted intense attention due to the unique morphology-dependent properties[32,33]. Hetero-structured nanosheets and nanorods have been applied in antibacterial therapy with improved bactericidal effect[34,35]. Compared with ordinary hetero-nanoparticles, Janus nanoparticles with separate domains make it possible to maximize the exploration of the distinct and synergistic properties of each component[36–40]. It is to be noted that Janus nanoparticles with photothermal compartments could generate a temperature gradient across the Janus boundary upon NIR light irradiation, thereby resulting in self-propelled active motion[40–44]. As a result, these nanomotors can achieve enhanced photothermal effect and tissue penetration depth. Photothermal bismuth selenide (BSe) nanosheets with excellent biocompatibility, metabolizability, and photothermal conversion efficiency demonstrate great potential in PTT and antibacterial therapy[45–47]. Therefore, if dextran and BSe nanosheets could be combined in Janus nanostructures, bacterial killing by efficient PTT could be realized through biofilm targeting and improved penetration. In addition, on-demand biofilm dispersion could offer an alternative approach for biofilm treatment.

Herein, NIR-responsive Janus nanoparticles integrating biocompatible dextran and photothermal nanoparticles are constructed by a facile strategy for biofilm-targeted advanced antibacterial therapy. As illustrated in Fig. 1, Janus-structured dextran-BSe (Dex-BSe) nanoparticles with BSe nanosheets positioned on one edge of the dextran nanospheres are synthesized. The dextran domain with the maximum exposure is anticipated to target the biofilm, enhance the penetration of nanoparticles, and facilitate the biofilm dispersion of Janus Dex-BSe, which can be employed in antibacterial wound dressings to remove biofilms efficiently. Interestingly, Janus Dex-BSe nanoparticles can realize enhanced dispersal of biofilms compared with dextran nanoparticles while the underlying molecular mechanisms are further elucidated by RNA-sequencing transcriptomics analysis. When NIR light irradiation is applied, the temperature elevation generated by BSe nanosheets endows the Janus nanoparticles with photothermal killing effect. More importantly, the self-propelled active motion induced by the unique Janus structure is envisioned to enhance penetration depth and PTT efficacy. Therefore, combining intrinsic EPS targeting of dextran domain and photothermal property of BSe nanosheets, Janus Dex-BSe nanoparticles exhibit synergistic enhancement of gene regulation for biofilm dispersion or active motion for photothermal killing, respectively. Moreover, positively charged Dex-BSe nanoparticles are designed to adhere to the bacterial surfaces through electrostatic interaction, thereby benefiting the efficiency of biofilm penetration, dispersion, and PTT. Janus-structured chitosan-BSe (CS-BSe) nanoparticles with comparable size and surface property are also prepared as the counterparts. The antibacterial activity of Janus Dex-BSe and CS-BSe nanoparticles against biofilms of *Staphylococcus aureus* (*S. aureus*) and methicillin-resistant *Staphylococcus aureus* (MRSA) are firstly investigated in vitro. Furthermore, the feasibility of selective application of Janus Dex-BSe nanoparticles to eradicate biofilms in vivo by dispersion or photothermal killing is validated by MRSA-infected mouse excisional wound model and abscess model, respectively.

## Results
### Preparation and characterization of Janus polysaccharide-bismuth selenide nanoparticles
BSe nanosheets were synthesized by a facile solution method with the assistance of poly(vinylpyrrolidone) (PVP). As shown in the transmission electron microscopy (TEM) image (Fig. 2a), BSe nanosheets with a nearly hexagonal morphology and an average size of ~90 nm were obtained. A few small BSe nanoparticles were observed on the nanosheets, which were probably formed by the random attachment of the nanoparticles and misorientation in the growing nanosheets[48]. The high-resolution TEM (HRTEM) image (inset of Fig. 2a) of the nanosheets shows continuous lattice fringes (0.208 nm) between the nanosheets and nanoparticles, which corresponds to the (110) plane of BSe, while the lattice spacing of 0.478 nm represents the (006) plane of BSe[49]. Aminated dextran (Dex-NH$_2$) was prepared by the reaction between activated hydroxyl groups of dextran and amino groups of ethylenediamine according to the previous report[21]. As shown in Supplementary Fig. 1, the typical proton nuclear magnetic resonance ($^1$H NMR) spectrum of Dex-NH$_2$ demonstrates that amino groups were successfully conjugated to the glucose units of dextran. Janus-structured Dex-BSe nanoparticles were synthesized employing a nonsolvent-aided counterion complexation strategy[50]. The nonsolvent ethanol was supposed to induce the counterion condensation of cationic Dex-NH$_2$, anionic ethylenediaminetetraacetic acid (EDTA) and negatively charged BSe nanosheets. As shown in Fig. 2b and Supplementary Fig. 2a, monodisperse Janus Dex-BSe nanoparticles with an average size of ~220 nm were prepared. BSe nanosheets were distributed on the outer edge of the dextran nanospheres. Statistical results indicate a high yield of 78.1% for Janus-structured Dex-BSe with a well-defined single BSe domain on a dextran nanosphere (Supplementary Fig. 2b). The formation of the Janus nanostructures might be related to the two-dimensional structure of BSe nanosheets and change of the interfacial energy between Dex-NH$_2$, BSe nanosheets, and the solvent[40,51]. By changing the type of polysaccharides from Dex-NH$_2$ to chitosan as the starting material, Janus-structured CS-BSe nanoparticles with comparable size of ~220 nm (Fig. 2c) were synthesized as the counterparts. The composition of Janus Dex-BSe was confirmed by the cross-sectional compositional line profiles and elemental mapping images (Fig. 2d). The C, O, and N elements were found to evenly distribute in the dextran domain, while the Bi and Se elements were located in the BSe domain, verifying the asymmetric Janus structure. Dynamic light scattering measurements confirm the comparable positively charged surface of Dex-BSe and CS-BSe nanoparticles (Supplementary Fig. 3). The positive surface charge is thought to facilitate nanoparticles interaction with bacterial cell membrane and promote penetration into biofilms[9]. The weight ratio of polysaccharides in CS-BSe and Dex-BSe was calculated to be ~60% by thermogravimetric analysis (TGA, Supplementary Fig. 4).

### Photothermal effect of Janus Dex-BSe nanoparticles
The intrinsic photothermal property of BSe nanosheets drove us to investigate the photothermal effect of Janus-structured Dex-BSe and CS-BSe nanoparticles. The temperature elevations of different concentrations of Dex-BSe aqueous solutions under 808 nm NIR laser irradiation were recorded. As shown in Fig. 2e, the temperature increased with the irradiation time and the Dex-BSe concentration. Accordingly, BSe nanosheets and CS-BSe nanoparticles with comparable BSe contents exhibit similar photothermal properties (Supplementary Fig. 5a, b). The photothermal conversion efficiency of Dex-

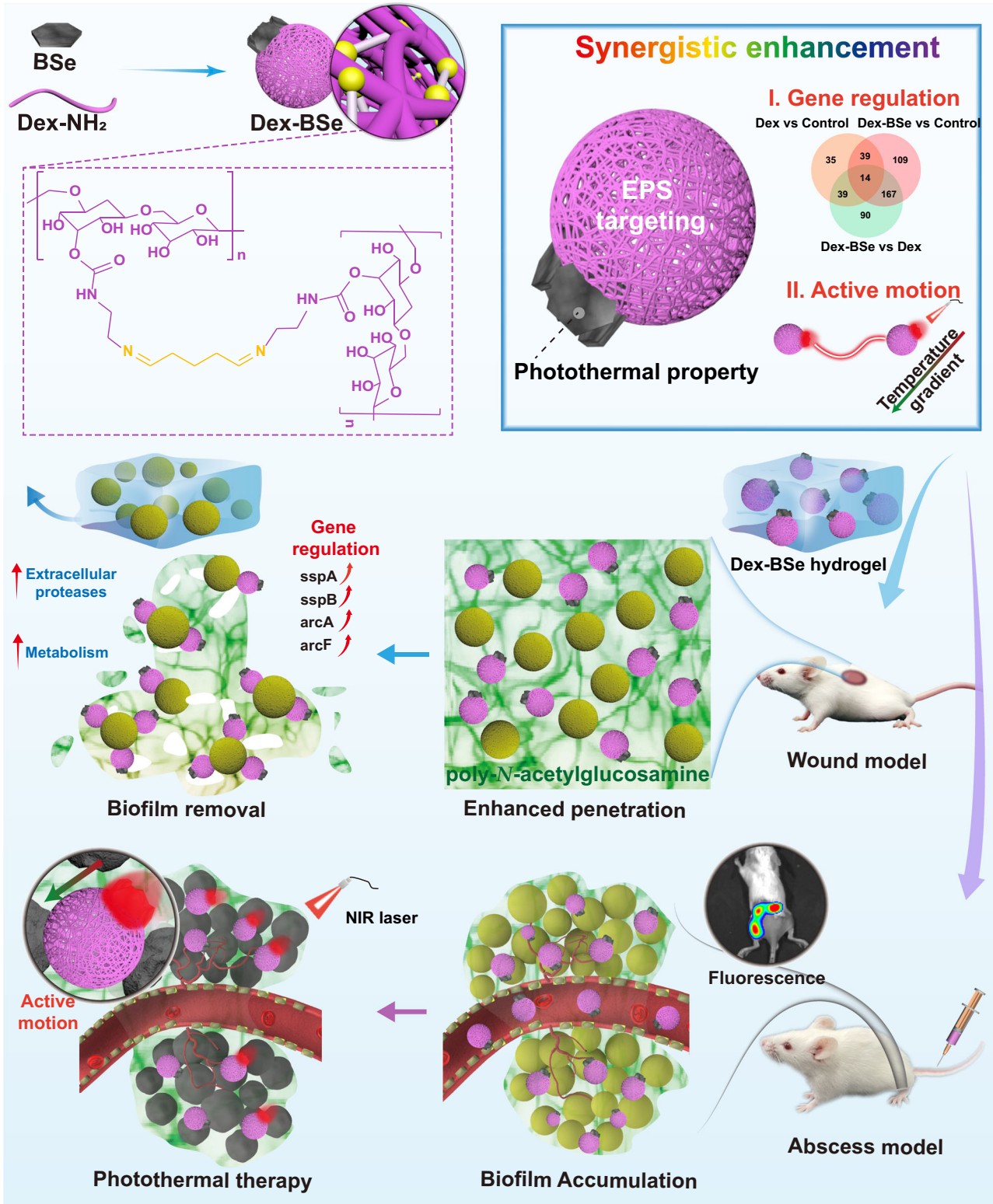

**Fig. 1 | Schematic illustration of Janus Dex-BSe nanoparticles for flexible elimination of biofilms.** Janus Dex-BSe nanoparticles are prepared by a nonsolvent-aided counterion complexation strategy. Benefiting from EPS targeting of dextran domain, Janus Dex-BSe nanoparticles could target and penetrate biofilms, further achieving synergistic effects between dextran and BSe components. As a result, efficient biofilm dispersion can be realized by enhanced gene up-regulation, which is employed in wound dressings to remove drug-resistant MRSA biofilm efficiently. Alternatively, biofilm accumulation can be realized after the intravenous injection of Janus nanoparticles. After NIR light irradiation is applied, enhanced photothermal effect induced by the active motion enables photothermal killing to eradicate the biofilm in the MRSA-infected abscess model.

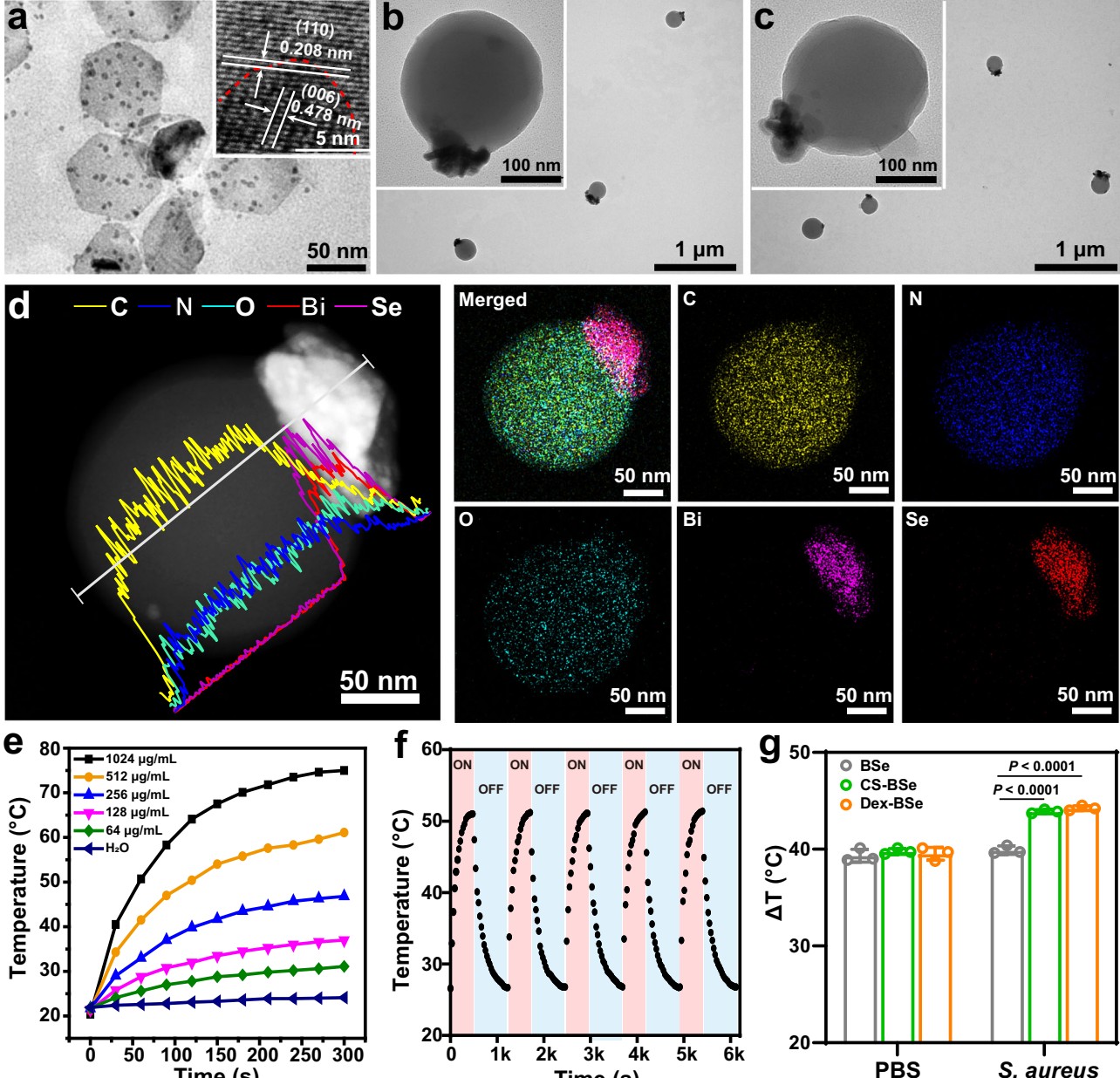

**Fig. 2 | Characterization of Janus-structured nanoparticles. a** TEM image and HRTEM (inset) image of BSe nanosheets. TEM images of Dex-BSe (**b**) and CS-BSe (**c**). **d** Scanning TEM image, corresponding cross-sectional compositional line profiles, and elemental mapping images of Dex-BSe. **a–d** Experiments were repeated three times independently with similar results. **e** Temperature elevation of Dex-BSe aqueous solutions with different concentrations under 808 nm laser irradiation (1.0 W/cm², 5 min). **f** Photothermal cycling curve of Dex-BSe (400 μg/mL) upon NIR laser irradiation. **g** The temperature variations of PBS and *S. aureus* (10⁵ CFU) culture media with BSe (205 μg/mL), CS-BSe (512 μg/mL) and Dex-BSe (512 μg/mL) under NIR laser, respectively. Data are presented as mean values±SD (*n* = 3 independent samples). Statistical significance was calculated by one-way ANOVA using the Tukey post-test. Source data are provided as a Source Data file.

BSe was determined to be ~31.5% by heating/cooling curve (Supplementary Fig. 5c), which was comparable with the previous reported photothermal conversion efficiency of BSe[46]. The excellent photothermal stability of Dex-BSe was further verified by the on-off cycling test (Fig. 2f). In addition, the morphology and size of Dex-BSe after NIR irradiation did not change noticeably (Supplementary Fig. 5d, e), verifying the stability of Dex-BSe for promising photothermal agents. More interestingly, the temperature elevation of Janus-structured Dex-BSe and CS-BSe in the medium containing *S. aureus* was significantly higher than that of BSe after 5 min of NIR irradiation (~44.0 °C vs ~39.9 °C, Fig. 2g). In contrast, BSe, Dex-BSe and CS-BSe exhibited similar temperature increases in PBS (~ 39.3 °C). These phenomena are consistent with enhanced photothermal effect of Janus-structured

nanomotors, which could be due to the conversion of kinetic energy into thermal energy after collision with the surrounding medium[40,43]. To verify the self-propelled active motion of Janus Dex-BSe nanomotors, the trajectories of Dex-BSe under NIR irradiation were recorded. As shown in Supplementary Fig. 6a and Supplementary Movie 1, Dex-BSe exhibited nondirectional Brownian motion in the absence of NIR irradiation. In contrast, the rapid propulsion of Dex-BSe was observed under NIR laser irradiation. Accordingly, the average velocity of the Dex-BSe increased significantly from 2.0 to 3.9 μm/s after NIR irradiation was applied (Supplementary Fig. 6b). The NIR-triggered active motion induced by temperature gradient across the Janus boundary contributed to the enhanced photothermal effect of Dex-BSe in the medium containing *S. aureus*. Taken together, Janus-structured

Dex-BSe nanoparticles are promising photothermal agents with satisfactory photothermal conversion efficiency, stability and enhanced photothermal effect with self-propelled active motion.

## Biofilm penetration and dispersion mediated by Janus Dex-BSe nanoparticles

The efficient penetration of nanoparticles into the dense protective layer of biofilms is prerequisite and crucial for the dispersion and photothermal eradication of biofilms[52,53]. To investigate the penetration and accumulation of Janus-structured Dex-BSe nanoparticles into mature biofilms, *S. aureus* biofilms were used as a model, while CS-BSe nanoparticles were employed as counterparts. After *S. aureus* biofilms were exposed to Cy5.5-labeled Dex-BSe and CS-BSe suspensions, *S. aureus* cells in the biofilm were stained by SYTO 9 for confocal laser scanning microscopy (CLSM) imaging. As shown in Fig. 3a, Dex-BSe with red fluorescence could penetrate quickly into the *S. aureus* biofilm within 10 min and accumulate in the biofilm in 30 min, indicating that Dex-BSe could diffuse throughout the biofilm efficiently. In contrast, limited penetration and diffusion of CS-BSe were observed after 60 min of incubation, as judged by negligible red fluorescence. These results indicate that Janus Dex-BSe nanoparticles could penetrate deep layers of biofilms due to the characteristic of dextran domain incorporation into the biofilm matrix via bacterial exoenzymes, which is consistent with previous reports that dextran-functionalized nanoparticles can target and penetrate into biofilms[19,22].

To verify enhanced biofilm penetration induced by the self-propelled active motion of Janus Dex-BSe, the biofilm penetration by Dex and Dex-BSe in the absence or presence of NIR light irradiation was investigated. As shown in Supplementary Fig. 7, limited penetration was observed for Dex nanoparticles with or without NIR irradiation. In contrast, Janus Dex-BSe quickly penetrated into the biofilm after NIR light irradiation was applied for 5 min while no noticeable penetration occurred in the absence of NIR light irradiation. These phenomena indicate that the enhanced penetration mediated by Janus Dex-BSe was attributed to the self-propelled action motion, which is consistent with previous works[18,54].

After the incubation time was extended to 2 h, *S. aureus* biofilms were found to be gradually dispersed by Janus Dex-BSe nanoparticles. As shown in Fig. 3b, the green fluorescent signals of bacteria in the biofilm treated with Dex-BSe decreased obviously from 4 to 6 h. The integrity of the biofilm was severely compromised and only partial biofilm fragments could be observed. In comparison, CS-BSe-treated biofilms remained intact after a prolonged period of 6 h, demonstrating thick biofilms (Fig. 3c). Meanwhile, the thickness of the biofilm incubated with Dex-BSe was evidently reduced after 6 h of incubation. The much fewer *S. aureus* colonies on the plate with the Dex-BSe group compared with the CS-BSe group confirmed the excellent ability of Dex-BSe to disperse mature biofilms (Fig. 3d). In addition, crystal violet-staining and dry weight assessment results indicate that Dex-BSe nanoparticles possessed high biofilm removal efficacy after 6 h of incubation (Fig. 3e, f and Supplementary Fig. 8). Both the considerable reduction in *S. aureus* colonies and biofilm biomass could be attributed to the contribution of the dextran domain in the Janus-structured Dex-BSe. In addition, scanning electron microscopy (SEM) imaging was performed to qualitatively analyze the cell density of biofilms after different treatments. As exhibited in Fig. 3g, Dex-BSe-treated biofilm with much fewer residual bacteria was apparently dispersed compared with the CS-BSe group. In addition, the dispersion effect on biofilms of Gram-negative *E. coli* mediated by Dex-BSe was further explored by crystal violet assay. As shown in Supplementary Fig. 9, the biofilm biomass exhibited negligible changes after treatment with different concentrations of Dex-BSe, indicating negligible dispersion effect on biofilms of Gram-negative bacteria, which is consistent with previous report[24]. Taken together, Dex-BSe nanoparticles with biofilm-targeting property demonstrated remarkable biofilm penetration and

dispersion capabilities against *S. aureus* biofilms. These results agree with the previous report that polymeric nanoparticles with dextran shell could diffuse into biofilms and attach to bacterial surfaces to induce bacterial detachment from biofilms[24].

## Mechanism of biofilm dispersion by Janus Dex-BSe nanoparticles

To verify the advantage of biofilm dispersion mediated by Janus Dex-BSe nanoparticles, the biofilm dispersion ability of BSe nanosheets, dextran nanoparticles (Dex,~220 nm), and the mixture of BSe nanosheets and Dex nanoparticles (Dex+BSe) were firstly evaluated by crystal violet-staining and standard plate counting method. As shown in Supplementary Fig. 10a–c, negligible decrease in biofilm biomass and *S. aureus* colonies was observed in the BSe and Dex+BSe groups. In contrast, Dex nanoparticles demonstrated obvious biofilm removal effect, verifying the important role of dextran shell in biofilm dispersion through enhanced solubility of the bacteria-nanoparticle complex. It is worth noting that Janus Dex-BSe induced significantly higher biofilm removal efficacy than Dex nanoparticles (Supplementary Fig. 10c). Meanwhile, a synergistic effect of biofilm dispersion between the Dex and BSe components in Janus Dex-BSe nanoparticles was achieved ($S>0$)[55]. Then, a wheat germ agglutinin-Alexa Fluor 488 (WGA-AF488) fluorescent conjugate was utilized to label the biofilm matrix due to its specific binding to poly-*N*-acetylglucosamine residues in the EPS of *S. aureus* biofilms[56]. Judging from the WGA staining assay (Supplementary Fig. 10d), Dex and Janus Dex-BSe nanoparticles could obviously reduce the EPS levels while Janus Dex-BSe nanoparticles demonstrated substantial higher removal effect on the EPS matrix, which is consistent with their excellent biofilm dispersion ability. In addition, CLSM was employed to visualize the distribution of different nanoparticles (labeled in red) in the biofilm matrix of EPS (in green). As shown in Fig. 4a and Supplementary Fig. 11, Janus Dex-BSe and Dex nanoparticles were colocalized with EPS while BSe and Dex+BSe group did not show overlap with EPS. The strong interaction between nanoparticles with dextran shells and EPS might contribute to the biofilm penetration and dispersion mediated by Janus Dex-BSe nanoparticles.

To further understand the underlying mechanism of the excellent biofilm dispersion by Janus Dex-BSe nanoparticles, RNA sequencing (RNA-seq) transcriptomics was performed. Gene expression profiles of biofilm cells treated with PBS, Dex, or Dex-BSe nanoparticles were analyzed. The differentially expressed genes (DEGs) among the three groups were displayed by the Venn diagram (Fig. 4b). Compared with the Dex group, 178 DEGs were upregulated and 132 DEGs were downregulated in the Dex-BSe group analyzed by volcano plot (Fig. 4c). In addition, a total of 329 genes including 223 upregulated DEGs and 106 downregulated DEGs were observed in the Dex-BSe group compared with the control group (Fig. 4c). In contrast, only an up-regulation of 87 DEGs and down-regulation of 40 DEGs were demonstrated in the Dex group in comparison with the control group (Fig. 4c). In other words, the difference in gene expression identified in the Dex-BSe group was more pronounced than in the Dex group when compared with the control group. Gene ontology (GO) analysis demonstrated that these DEGs between the Dex-BSe group and Dex group were associated with biological processes, cellular components, and molecular functions (Fig. 4d).

In addition, enrichment analysis of the Kyoto Encyclopedia of Genes and Genomes (KEGG) pathways showed that the DEGs between the Dex-BSe group and Dex group were enriched in amino acids synthesis and metabolism-related pathways, such as arginine biosynthesis, D-amino acid metabolism, as well as alanine, aspartate and glutamate metabolism (Fig. 4e), which are correlated with biofilm formation and biofilm detachment[57–59]. Moreover, the comparison between the Dex-BSe group and control group demonstrated that in addition to amino acids synthesis and metabolism-related pathways,

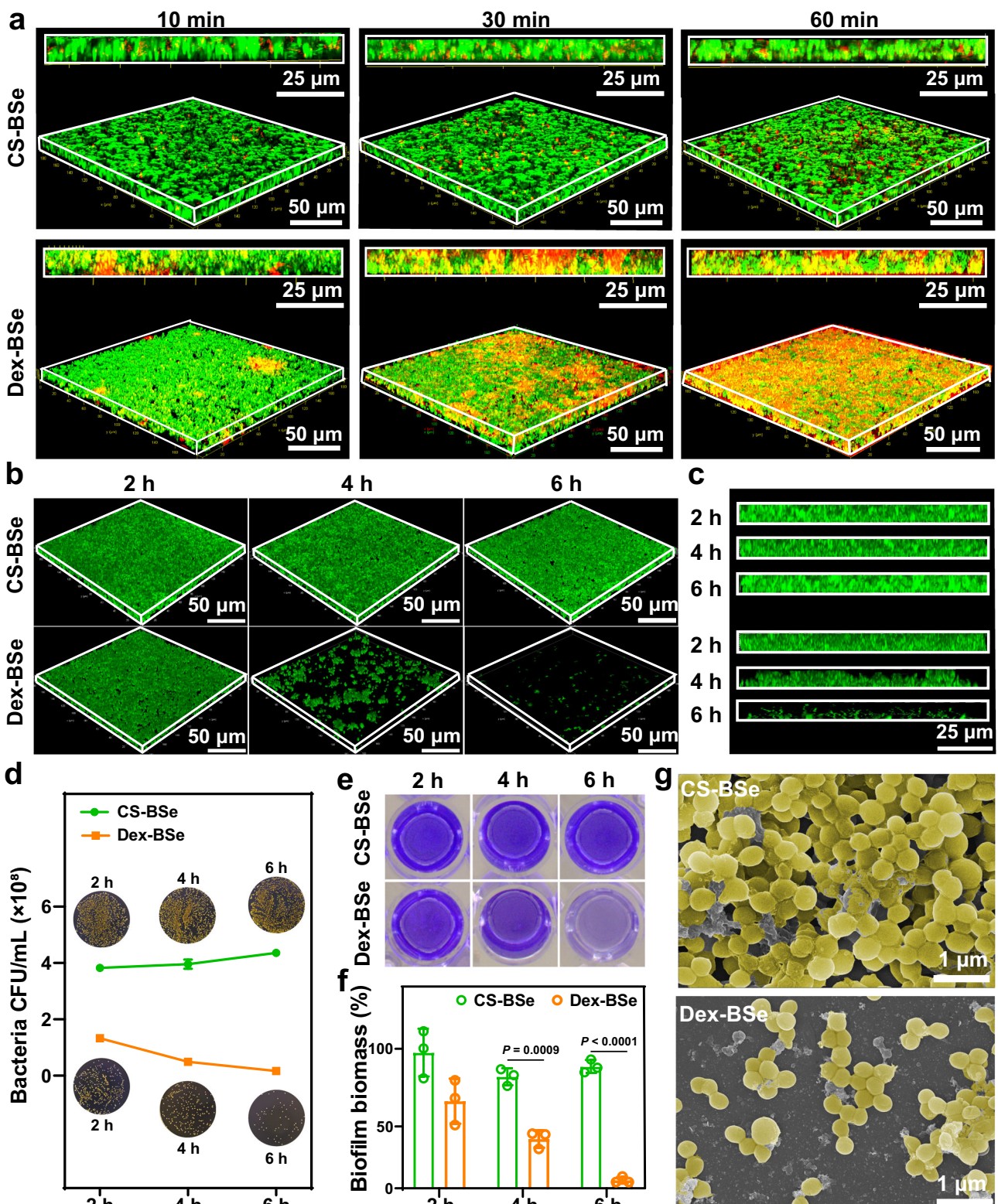

**Fig. 3 | Biofilm penetration and dispersion by nanoparticles. a** 3D CLSM images and corresponding z-stack images of *S. aureus* biofilms treated with Cy5.5-labeled Dex-BSe and CS-BSe for 10, 30, and 60 min. Green: live bacteria, Red: Cy5.5-labeled nanoparticles. 3D CLSM images (**b**) and corresponding z-stack images (**c**) of *S. aureus* biofilms treated with CS-BSe and Dex-BSe for 2, 4, and 6 h. Green: live bacteria. **d** Bacterial counts in *S. aureus* biofilms treated with CS-BSe and Dex-BSe for 2, 4, and 6 h, with photographs of bacterial cultures shown inset. Data are presented as mean values±SD (*n* = 3 independent samples). **e** Pictures of biofilms stained by crystal violet. **f** Quantitative analysis of the crystal violet-stained biofilms treated with CS-BSe and Dex-BSe at different incubation times. Data are presented as mean values±SD (*n* = 3 independent samples). Statistical significance was calculated by two-tailed Student's *t*-test. **g** SEM images of *S. aureus* biofilms treated with CS-BSe and Dex-BSe for 6 h. Source data are provided as a Source Data file.

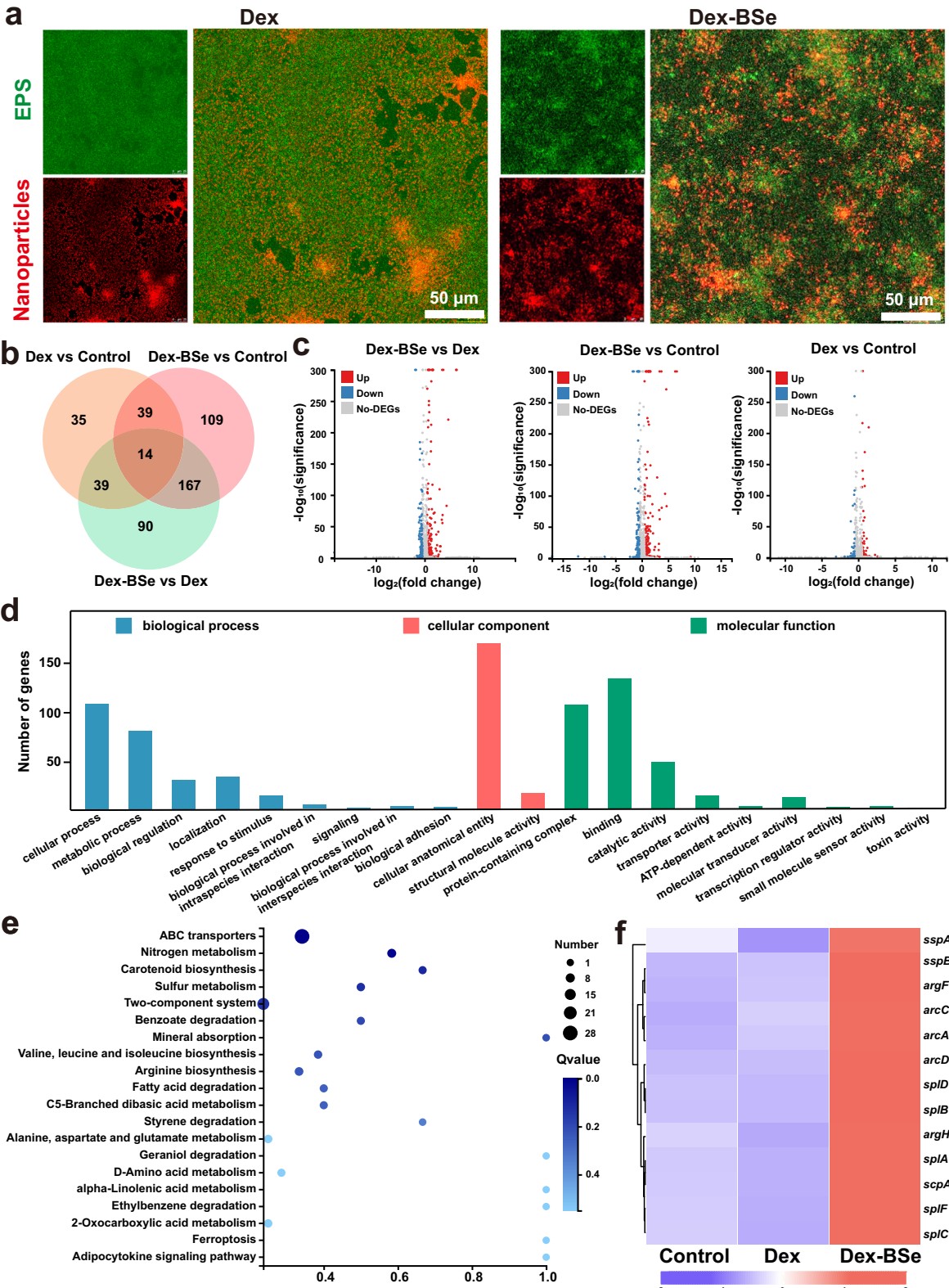

**Fig. 4 | Mechanism of biofilm dispersion mediated by Janus Dex-BSe nanoparticles. a** CLSM images of *S. aureus* biofilms treated with Dex and Dex-BSe nanoparticles for 60 min. Green: EPS, Red: Cy5.5-labeled nanoparticles. Experiments were repeated three times independently with similar results. **b** Venn diagram of DEGs in the three groups. **c** Volcano plot of DEGs in biofilms after different treatments. **d** GO annotation analysis of DEGs in biofilms treated with Dex and Dex-BSe. **e** KEGG enrichment analysis of DEGs in biofilms after treatment with Dex and Dex-BSe (512 μg/mL). **f** Heatmap of DEGs in biofilm-related pathways.

DEGs were also associated with the quorum sensing and purine metabolism (Supplementary Fig. 12a), which play important roles in *S. aureus* biofilm dispersion[57,60]. On the other hand, DEGs between the Dex group and control group were only associated with the quorum sensing and purine metabolism (Supplementary Fig. 12b). These results indicate that Janus Dex-BSe nanoparticles could affect more biofilm-related pathways than Dex, which may be responsible for their excellent biofilm dispersion efficacy. As shown in Fig. 4f, hierarchical clustering heatmap of biofilm-related DEGs demonstrates that the genes encoding extracellular proteases, such as *sspA* and *sspB*, were significantly upregulated in the Dex-BSe group compared with the control and Dex groups, which are involved in biofilm detachment[60]. In addition, the genes involved in arginine biosynthesis and arginine catabolism such as *argF*, *argH*, *arcC*, *arcA*, and *arcD* were also substantially upregulated after Dex-BSe treatment, which is supposed to inhibit the formation of biofilm and facilitate biofilm dissociation[57]. RT-PCR assay was further performed to analyze the expression levels of typical genes, verifying the RNA-seq results (Supplementary Fig. 13). To investigate the effect of BSe on biofilm-related genes, DEGs between the BSe group and the control group were analyzed. It was found that 123 DEGs were upregulated and 201 DEGs were downregulated in the BSe group compared with the control group (Supplementary Fig. 14a). KEGG enrichment analysis demonstrates that DEGs between the BSe group and control group were related to quorum sensing (Supplementary Fig. 14b), which may contribute to the biofilm dispersion mediated by Dex-BSe. Collectively, it is speculated that Janus-structured Dex-BSe nanoparticles are favorable for biofilm dispersion through the up-regulation of genes related to extracellular proteases, amino acids synthesis and metabolism.

## Photothermal antibiofilm activities of Janus Dex-BSe nanoparticles

Encouraged by the satisfactory photothermal properties of Janus-structured nanoparticles, the broad-spectrum bacterial killing effect by Janus Dex-BSe nanoparticles via PTT was investigated employing Gram-positive *S. aureus* and Gram-negative *Escherichia coli* (*E. coli*). As shown in Fig. 5a, b, colony counting method was utilized to quantitatively examine the antibacterial activity of nanoparticles against *S. aureus*. The killing effects of BSe, CS-BSe, and Dex-BSe nanoparticles were found to be concentration-dependent under NIR light irradiation. At the BSe concentration of 205 μg/mL, nearly 100% bacterial death were induced in the Dex-BSe and CS-BSe groups, while only ~30% bacteria were killed in the BSe group. The excellent photothermal killing activities of Dex-BSe and CS-BSe against *S. aureus* could be attributed to their enhanced photothermal effect in bacterial culture medium and surface properties derived from the Janus structure (Fig. 2g and Supplementary Fig. 3). Positively charged Dex-BSe and CS-BSe nanoparticles were believed to facilitate adhesion with the negative bacteria through electrostatic interactions[52,53]. The interaction between *S. aureus* and nanoparticles was visualized by SEM imaging. As displayed in Supplementary Fig. 15a–c, compared with BSe nanoparticles, obviously more CS-BSe and Dex-BSe adhered to the surface

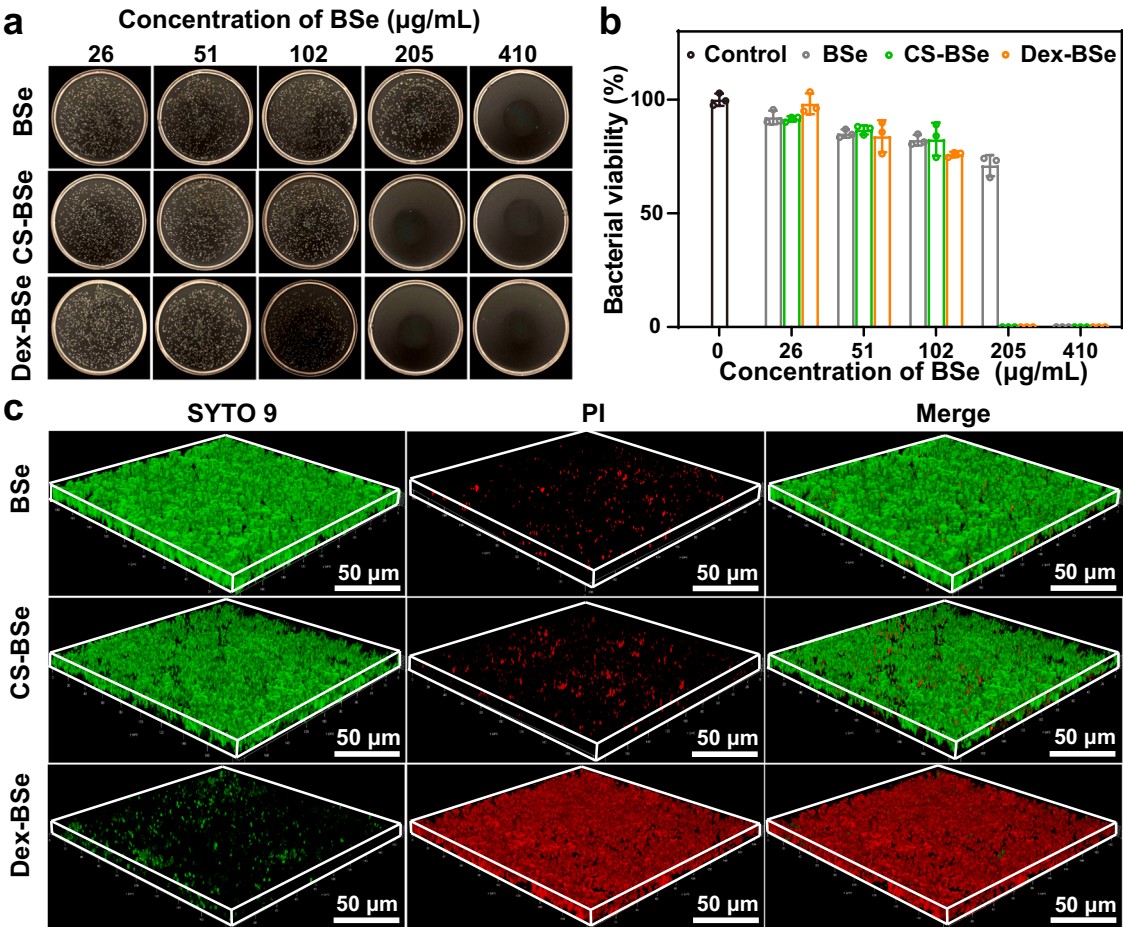

**Fig. 5 | Bacterial killing and antibiofilm effects of Dex-BSe nanoparticles.** Photographs of *S. aureus* colonies (**a**) and the corresponding bacterial viability (**b**) treated with BSe, CS-BSe and Dex-BSe, respectively under 808 nm NIR light irradiaiton (1.0 W/cm², 5 min). Data are presented as mean values ± SD (*n* = 3 independent samples). **c** 3D CLSM images of live/dead staining after *S. aureus* biofilms treated with BSe, CS-BSe, and Dex-BSe at BSe concentration of 205 μg/mL, respectively under 808 NIR light (1.0 W/cm², 5 min) irradiaiton. Green: live bacteria, Red: dead bacteria. Source data are provided as a Source Data file.

of *S. aureus*, confirming the enhanced affinity toward bacteria. In addition, zeta potential analysis of *S. aureus* treated with nanoparticles revealed that positively charged Dex-BSe and CS-BSe nanoparticles enhanced the surface charge of bacteria from negative to positive as the nanoparticle concentration increased (Supplementary Fig. 15d). In contrast, negligible changes were found in the zeta potential of bacteria incubated with different concentrations of BSe. The antibacterial activity of Dex-BSe and CS-BSe against *E. coli* further demonstrated their broad-spectrum photothermal killing effect (Supplementary Fig. 16).

Motivated by the enhanced biofilm penetration (Fig. 3a) and photothermal killing effects (Fig. 5a, b) of Dex-BSe nanoparticles, their antibiofilm efficacy was then explored by live/dead staining assay. Specifically, *S. aureus* biofilms were incubated with BSe, CS-BSe, and Dex-BSe suspensions for 10 min, respectively, and then exposed to NIR light irradiation for 5 min. Subsequently, SYTO 9/propidium iodide (PI) staining was performed to visualize live bacteria with green fluorescence and dead bacteria with red fluorescence. As shown in Fig. 5c, the Dex-BSe group displayed apparent red fluorescence with slight green fluorescence spots throughout the whole biofilm, indicating excellent photothermal bacterial killing and effective biofilm eradication. On the other hand, biofilms treated with BSe and CS-BSe exhibited only a small amount of red fluorescence. In addition, the quantitative analysis employing crystal violet-staining and standard plate counting method confirmed the substantially higher antibiofilm effect of Dex-BSe than BSe and CS-BSe nanoparticles (Supplementary Fig 17). The insignificant biofilm inhibition mediated by BSe and CS-BSe was supposed to be caused by limited penetration of biofilms by these two nanoparticles. To verify the effect of penetration on killing effect, the photothermal killing ability of Dex-BSe against *S. aureus* biofilm after different incubation times was further investigated by standard plate counting method. As shown in Supplementary Fig. 18, significant photothermal killing effect of Dex-BSe against *S. aureus* biofilm was observed after 10 min of incubation. As the incubation time was extended to 60 min, the bacterial viability was further reduced, indicating the enhanced antibiofilm efficacy under NIR light irradiation as the penetration of Janus Dex-BSe in *S. aureus* biofilm was increased. Taken together, Dex-BSe with excellent biofilm penetration and bacterial killing effects demonstrated remarkable photothermal elimination of *S. aureus* biofilms.

### Dispersion and photothermal killing effect of Janus Dex-BSe nanoparticles against drug-resistant biofilms

To verify the efficacy of Janus Dex-BSe nanoparticles in the treatment of drug-resistant biofilms, methicillin-resistant *Staphylococcus aureus* (MRSA) biofilms were employed as a proof-of-concept biofilm model. As expected, Janus Dex-BSe nanoparticles could disperse the MRSA biofilm gradually (Fig. 6a). With the extension of incubation time, the integrity of the MRSA biofilm was disrupted obviously and only partial biofilm fragments could be observed after 6 h. Meanwhile, there were noticeable changes in the thickness of the biofilm after the treatment with Dex-BSe. In contrast, no dispersal of biofilms exposed to BSe and CS-BSe was observed, implying that only Dex-BSe could weaken the attachment of the bacteria to the biofilm and induce excellent biofilm removal. The significant reductions in the biofilm bacterial cell counts in the Dex-BSe group after 4 and 6 h of treatment further confirmed the dispersion efficiency of Dex-BSe against drug-resistant MRSA biofilms (Fig. 6b). For BSe and CS-BSe groups, no noticeable biofilm reduction was observed after the incubation time was extended to 6 h.

To evaluate the photothermal killing ability of Janus Dex-BSe on MRSA biofilms, biofilms stained by SYTO 9 and PI were visualized by CLSM imaging. As shown in Fig. 6c, biofilms incubated with Dex-BSe under NIR light irradiation displayed strong red fluorescence. Meanwhile, the BSe and CS-BSe groups exhibited only slight red fluorescent spots on the biofilm, indicating no obvious antibiofilm effect. In addition, z-stack images of MRSA biofilms after different treatments revealed that only Dex-BSe could penetrate deep layers of biofilms to exert photothermal killing effect. Quantitative results show that significant reduced bacterial viability (~4%) was found in the Dex-BSe group compared with the other groups, verifying their excellent efficacy to eliminate MRSA biofilms (Fig. 6d). In contrast, CS-BSe and BSe only showed a certain degree of killing effect on MRSA biofilms. Moreover, the appreciable stronger antibiofilm activity of CS-BSe than BSe may be attributed to the active motion induced by the Janus structure, which in turn leads to enhanced photothermal effect and biofilm penetration. Taken together, satisfactory penetration, dispersion, and photothermal eradication of MRSA biofilms render Janus Dex-BSe nanoparticles promising for the treatment of drug-resistant biofilm-associated infections in vivo.

### Biocompatibility of Dex-BSe nanoparticles in vitro

Biocompatibility of antibacterial materials is essential for their biomedical applications in vivo. Antibacterial nanoparticles are expected to possess compromised cytotoxicity and excellent hemocompatibility while maintaining potent antibacterial activity against bacteria. The cytotoxicity of Dex-BSe was studied by methylthiazolyl tetrazolium (MTT) assay while L929 cells were selected as the model normal cells. As exhibited in Supplementary Fig. 19a, BSe, CS-BSe, and Dex-BSe nanoparticles did not noticeably affect the cell viability of L929 cells (>80%) at BSe concentrations of 26–410 μg/mL, indicating favorable biocompatibility at the cellular level.

Additionally, the hemolytic activity of Janus Dex-BSe on red blood cells (RBCs) was further investigated (Supplementary Fig. 19b). The transparent and colorless RBC supernatants were observed after the treatments with Janus Dex-BSe. The corresponding hemolysis ratio was much lower than the permissible limit (5%), confirming good hemocompatibility in the wide concentration rage of 64–2048 μg/mL. Taken together, the favorable biocompatibility of Janus Dex-BSe nanoparticles ensures their suitability for anti-infective therapy in vivo.

### Dispersal of biofilms by Janus Dex-BSe nanoparticles in vivo

Encouraged by the excellent biocompatibility and attractive antibiofilm activity of Dex-BSe against drug-resistant biofilms in vitro, the biofilm dispersal efficacy of Dex-BSe in vivo was evaluated by an MRSA-infected mouse excisional wound model. To establish the infection model, 10 μL of $10^8$ CFU/mL bacteria was inoculated at the wound site and incubated for 24 h to induce infected wound with MRSA biofilms (Fig. 7a). Hydrogel dressings with a diameter of ~5 mm were prepared containing CS-BSe and Dex-BSe nanoparticles at a BSe concentration of 2 mg/mL, respectively (Fig. 7b). The wound was then covered with a hydrogel dressing every 6 h for a total of three treatments (Fig. 7c). Hydrogel containing PBS were utilized as infection control. Subsequently, the number of bacteria on the wounds was quantified by standard plate counting method, in which wound tissue was harvested after treatment and placed on agar plates. As shown in Fig. 7d, the treatment with Dex-BSe could effectively reduce biofilm bacteria at the wound site. Based on the bacterial counting results, the number of bacterial colonies observed in the Dex-BSe group was significantly lower than that in the CS-BSe group, indicating the excellent biofilm dispersion effect of Dex-BSe (Fig. 7e). Taken together, benefiting from the strong interaction between the dextran domain of the Janus nanoparticles and EPS (Fig. 4a), as well as the significant up-regulation of biofilm-related genes (Fig. 4f), Janus Dex-BSe nanoparticles were demonstrated to disperse drug-resistant MRSA biofilms effectively in vivo. The encouraging results also verified the hypothesis that Janus Dex-BSe nanoparticles could be used for antibacterial wound dressings and disinfectant rinses by bacterial debridement.

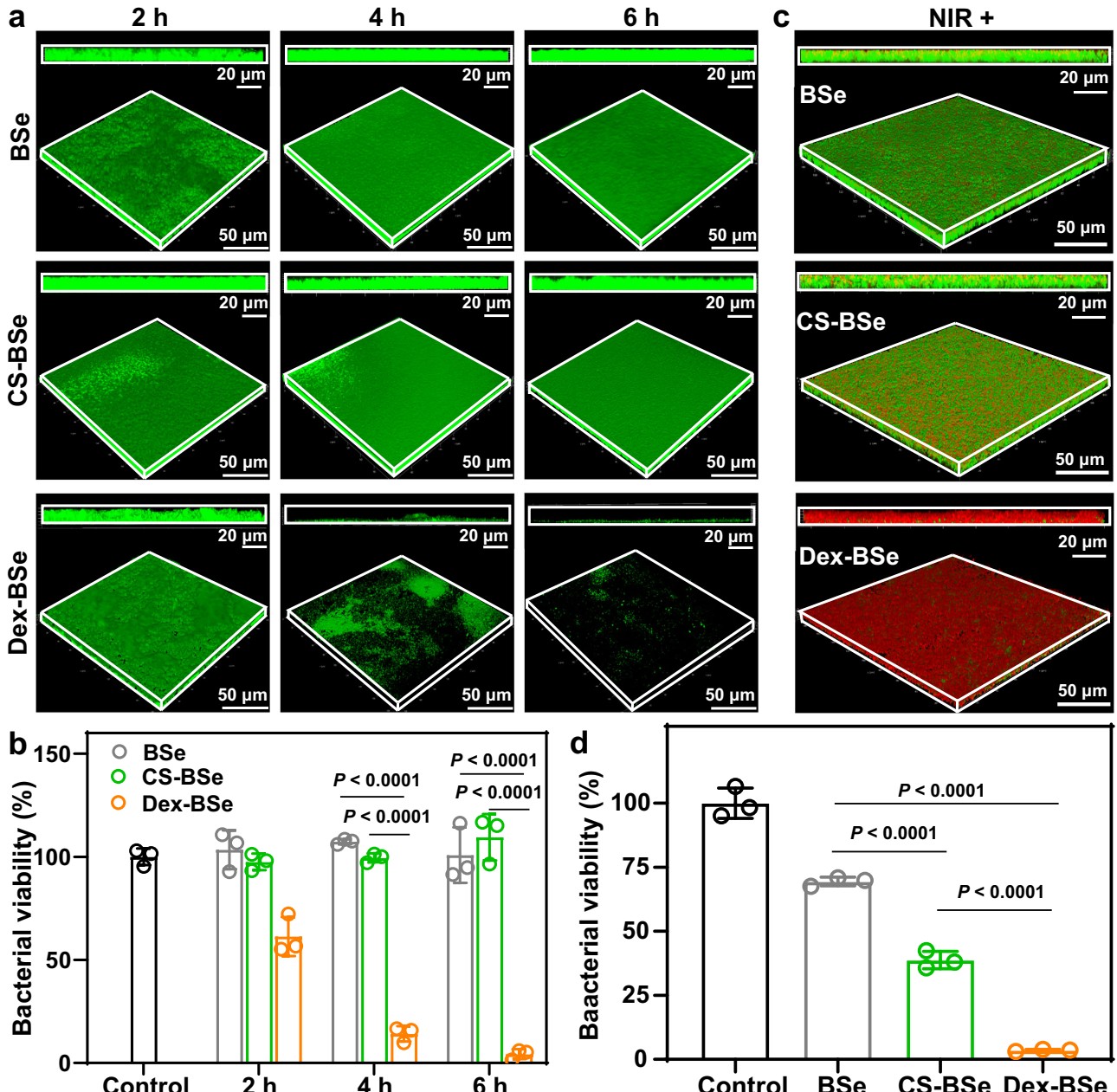

**Fig. 6 | Biofilm dispersion and photothermal killing effect against MRSA biofilms. a** 3D CLSM images and corresponding z-stack images of MRSA biofilms treated with BSe, CS-BSe, and Dex-BSe for 2 h, 4 h, and 6 h, respectively. Green: live bacteria. **b** Corresponding viability of MRSA biofilms after different incubation time determined by a typical plate counting method in (**a**). Data are presented as mean values±SD (*n* = 3 independent samples). Statistical significance was calculated by one-way ANOVA using the Tukey post-test. **c** 3D CLSM images of live/dead staining after MRSA biofilms treated with BSe, CS-BSe and Dex-BSe, respectively under 808 NIR light (1.0 W/cm², 5 min) irradiaiton. Green: live bacteria, Red: dead bacteria. **d** Corresponding viability of MRSA biofilms after different treatments. Data are presented as mean values ± SD (*n* = 3 independent samples). Statistical significance was calculated by one-way ANOVA using the Tukey post-test. Source data are provided as a Source Data file.

## Photothermal antibiofilm performance of Janus Dex-BSe nanoparticles in vivo

Considering the desirable photothermal antibiofilm activity of Dex-BSe against MRSA in vitro, it was expected that Janus Dex-BSe could be used to eradicate drug-resistant biofilms in vivo. The abscess model was established by injecting MRSA into the left rear thigh muscle of mice (Fig. 8a). Inspired by the biofilm-penetration and targeting ability of dextran-functionalized nanoparticles[19,22], the in vivo accumulation of Dex-BSe nanoparticles at the site of infection was firstly investigated prior to treatment. Cy5.5-labeled Dex-BSe and CS-BSe nanoparticles were monitored by real-time fluorescence imaging. As displayed in

Fig. 8b, a time-dependent accumulation behavior at the infected abscess was observed following intravenous injection of Janus Dex-BSe nanoparticles (10 mg/kg). The fluorescence signal increased gradually over time and then decreased slowly. However, the fluorescence of CS-BSe could only be detected in normal tissues. More importantly, obviously strong and persistent fluorescence at the infection site was found in the Dex-BSe group during the 24 h post-injection period, verifying the enhanced penetration and retention of Dex-BSe at the infection site. Notably, the fluorescence intensity reached a maximum at 8 h post-injection, providing valuable information for the realization of imaging-guided treatment. Additionally, the distribution of

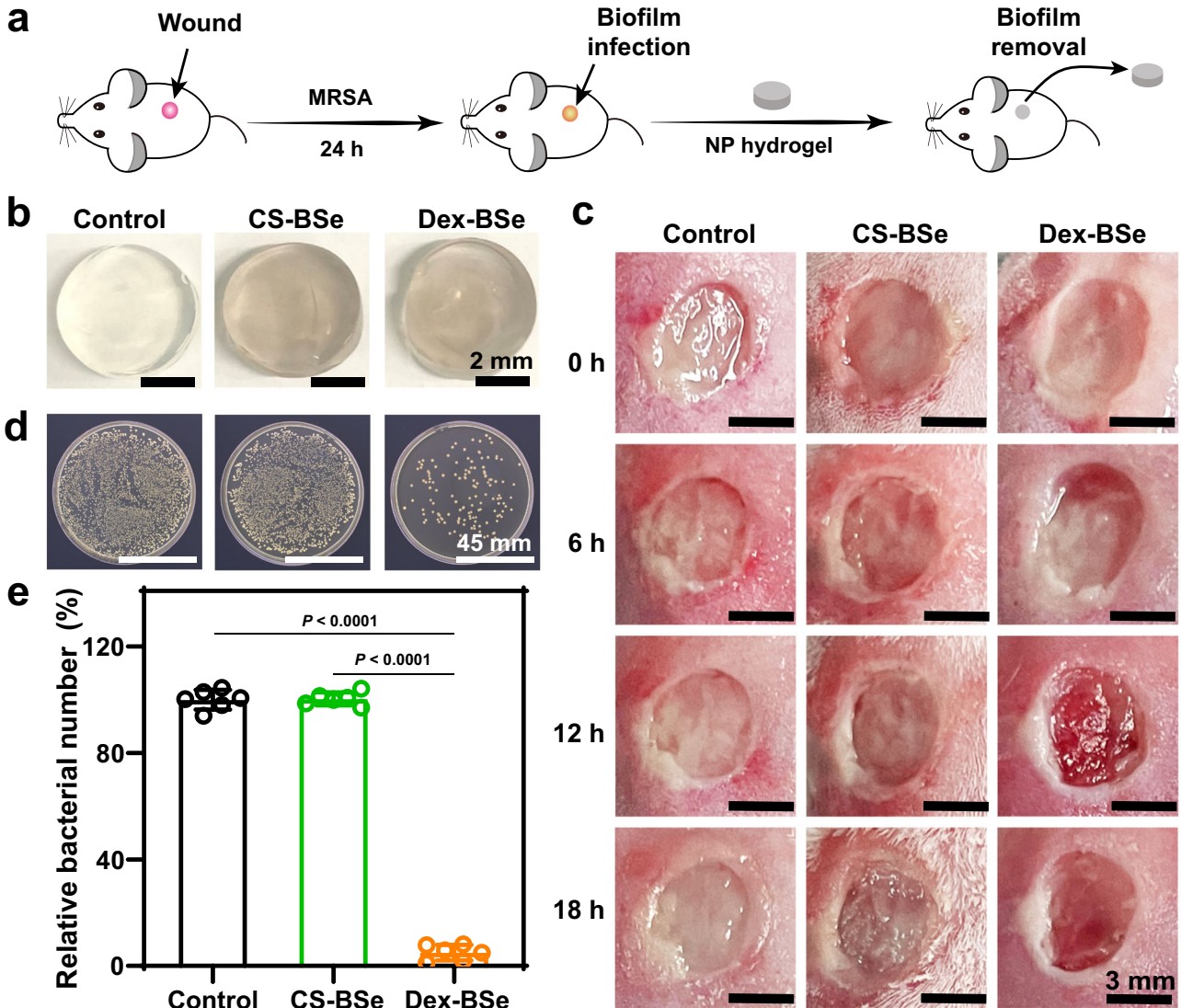

**Fig. 7 | Dispersal of biofilms by nanoparticles in vivo. a** Scheme of in vivo study of biofilm dispersion ability of the hydrogel with nanoparticles (NP hydrogel) against MRSA biofilm in established murine excision wound model. **b** Photographs of hydrogels containing CS-BSe and Dex-BSe nanoparticles, respectively. **c** Time-dependent photographs of wounds with different treatments. Photographys (**d**) and quantification analysis (**e**) of bacterial colonies from wound tissues of different groups after 18 h. Data are presented as mean values±SD ($n = 6$ independent samples). Statistical significance was calculated by one-way ANOVA using the Tukey post-test. Source data are provided as a Source Data file.

nanoparticles in main organs and infected abscesses was visualized at 8 h post-injection, further suggesting a higher accumulation of Dex-BSe in the MRSA biofilm-infected abscess (Fig. 8c). Considerable nanoparticle was also found in liver at this stage, which is consistent with previous reports that metabolizable BSe nanoparticles were mainly cleared by the liver[49,61]. All these results confirmed the high biofilm penetration and targeting of Dex-BSe nanoparticles, which guarantees their promising photothermal killing effect.

Motivated by the enhanced photothermal effect in vitro derived from the Janus structure and the desirable accumulation at the infection site benefiting from the dextran component, the photothermal effect of Dex-BSe nanoparticles in vivo was then explored. CS-BSe and Dex-BSe nanoparticles were intravenously injected into MRSA-infected abscesses. The infection sites were exposed to an 808 NIR irradiation at 8 h after intravenous injection, which was the optimal time point selected according to the results of fluorescence imaging (Fig. 8b). Infrared thermal images and temperature changes of the infection site were recorded every 30 s. As exhibited in Fig. 8d, e, after 5 min of light irradiation, distinct color change and temperature elevation (~24 °C)

were found in the abscess of Dex-BSe-treated mice. In contrast, only slight changes were detected in the CS-BSe group, underlining the great potential of Janus Dex-BSe for PTT in vivo. At the same time, the feasibility of imaging-guided treatment of infection in vivo was verified.

To investigate the photothermal biofilm eradiation performance of Janus Dex-BSe in vivo, the MRSA-infected mice were randomly divided into three groups with six mice in each group. Subsequently, PBS, CS-BSe and Dex-BSe were intravenously administrated. 808 nm NIR light irradiation was performed at 8 h post-injection. After 7 days of treatment, the mice were sacrificed and infected tissue was collected for evaluation. Plate counting method was utilized to quantify the bacterial counts in infected abscesses. As displayed in Fig. 8f, CS-BSe was found to be ineffective against biofilm infection. In contrast, the number of bacteria in the Dex-BSe group was significantly less than the other groups, confirming the remarkable performance of Dex-BSe for photothermal eradication of biofilms. Furthermore, hematoxylin and eosin (H&E) staining was applied to investigate the antibiofilm effect of Dex-BSe. As shown in Fig. 8g, obvious inflammatory cells were

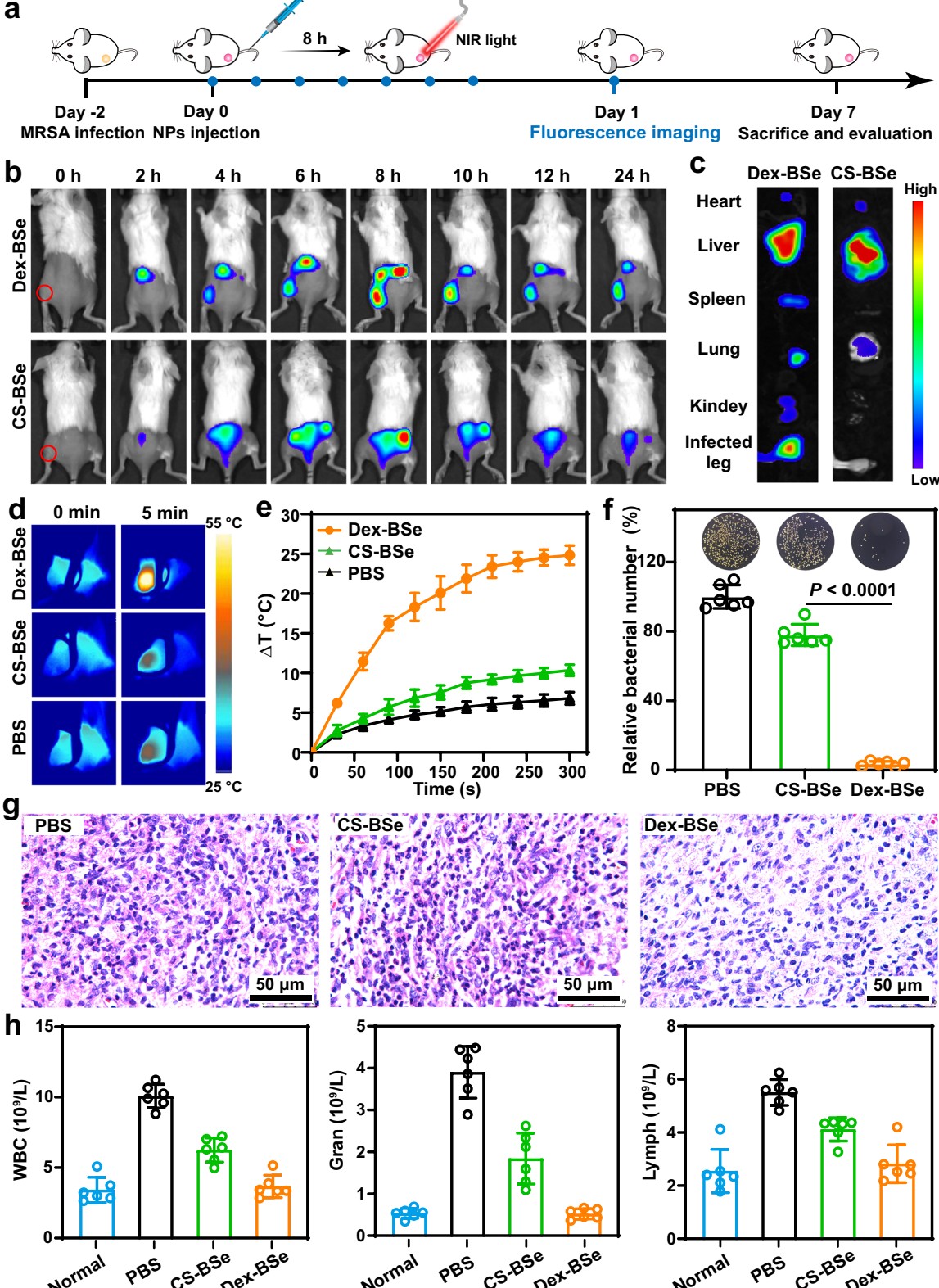

**Fig. 8 | Photothermal killing effect of nanoparticles in vivo. a** Timeline for MRSA infection model establishment, monitoring of abscess by imaging, and systemic treatment. **b** Time-dependent fluorescence images of MRSA biofilm-infected mice after intravenous injection of CS-BSe and Dex-BSe, respectively. **c** Florescence images of major organs and infected abscess after different treatments at 8 h post-injection. Photothermal images (**d**) and temperature elevation curves (**e**) of mice treated with CS-BSe and Dex-BSe by intravenous injection under 808 nm irradiation. Data are presented as mean values±SD (*n* = 4 independent samples). **f** Photographs and quantitative analysis of bacterial colonies in abscesses of different treatment groups after 7 days. Data are presented as mean values±SD (*n* = 6 independent samples). Statistical significance was calculated by two-tailed Student's *t*-test. **g** H&E staining images of infected tissues from different treatment groups after 7 days. **h** Blood biochemical indexes of mice after 7 days of different treatments. Data are presented as mean values±SD (*n* = 6 independent samples). Source data are provided as a Source Data file.

observed in the PBS and CS-BSe groups, suggesting severe bacterial infection. As expected, negligible inflammatory cells were found in the Dex-BSe group, further confirming the advantage of Janus Dex-BSe in terms of effective accumulation and photothermal killing effect at the infection site.

## Safety assessment of Dex-BSe nanoparticles in vivo

Moreover, the blood samples of the mice after 7 days of treatment were tested and analyzed. As shown in Fig. 8h, compared with the control and CS-BSe groups, blood biochemical indexes such as white blood cells (WBC), neutrophilic granulocytes (Gran), and lymphocytes (Lymph) in the Dex-BSe group returned to normal after treatment, indicating the effective inhibition of the biofilm infection. Meanwhile, levels of red blood cells, hemoglobin, and hematocrit were within normal ranges (Supplementary Fig. 20). Histological analysis of major organs including heart, liver, spleen, lung, and kidney performed after 7 days of treatment showed no noticeable damages or inflammatory lesions (Supplementary Fig. 21). In addition, a slight increase in the body weight of the mice was observed during the treatment period (Supplementary Fig. 22). These results validate the ideal biosafety and biocompatibility of Janus Dex-BSe nanoparticles. Collectively, Janus Dex-BSe nanoparticles are promising safe and efficient candidates for photothermal eradication of biofilm infections in vivo.

In addition, the dose-dependent toxicity and potential side effects of the Janus Dex-BSe nanoparticles were then evaluated. No obvious difference in body weight was observed between the control group and Dex-BSe-treated groups with different doses (3, 10, and 50 mg/kg Dex-BSe, Supplementary Fig. 23), indicating Dex-BSe nanoparticles have no evident adverse effects on mice. Oxidative stress has been considered as a major concern for nanoparticle-induced neurotoxicity[62,63]. To investigate the toxicity of different doses of Dex-BSe on the central nervous system of mice, the activity of superoxide dismutase (SOD), glutathione (GSH), and the malondialdehyde (MDA) level in the central nervous system were measured, which are closely correlated with oxidative stress. As shown in Supplementary Fig. 24, the GSH and SOD activity, as well as the MDA level of hippocampus and cerebral cortex in each dose group after 7 days of treatment did not show obvious changes compared with the control group, indicating no noticeable oxidative injuries was induced. Moreover, no obvious damage to neurons was observed in the hippocampus (CA1, CA3, and dentate gyrus (DG) regions) and cerebral cortex after different treatments (Supplementary Fig. 25), confirming compromised toxicity of Dex-BSe to the central nervous system.

In order to evaluate the dose-dependent toxicity and potential effects on lung, bronchoalveolar lavage fluid (BALF) after the treatment of different doses of Dex-BSe was collected for the detection of pro-inflammatory cytokines, which could be induced by cellular damage via oxidative stress[64,65]. As shown in Supplementary Fig. 26, compared with the control group, the levels of pro-inflammatory cytokines including tumor necrosis factor-alpha (TNF-α), interleukin-6 (IL-6), and interleukin-1β (IL-1β) were found to increase significantly in BALF of mice treated with high dose of Dex-BSe (50 mg/kg), while no obvious changes were observed in the low and medium dose groups (3 and 10 mg/kg of Dex-BSe), demonstrating a dose-dependent oxidative damage. Histological analysis of the lungs indicates negligible lung damage or fibrosis (Supplementary Fig. 27). However, pro-inflammatory cells infiltration was observed in the high-dose group while negligible inflammation was found in the low and medium groups, verifying the pulmonary inflammation induced by high dose of Dex-BSe via oxidative stress. No significant inflammation-associated spleen enlargement was detected in the low and medium doses group, while obvious enlargement and a significant increase of spleen index were observed in the high dose treatment group (Supplementary Fig. 28), implying the possible spleen toxicity caused by high dose of Dex-BSe. Additionally, markers for hepatic (alanine aminotransferase (ALT) and aspartate aminotransferase (AST)) and renal function (creatinine (CREA) and urea nitrogen (UREA)) were studied to evaluate the toxicity of Dex-BSe. As displayed in Supplementary Fig. 29, no obvious changes in ALT, AST, CREA, and UREA were observed after 7 days of treatments compared with the control group, indicating the normal liver and kidney function of mice after the treatment with different doses of Dex-BSe. Collectively, Dex-BSe nanoparticles may not cause apparent damage to the central nervous system, liver or kidneys while high doses of Dex-BSe nanoparticles may induce potential pulmonary and splenic toxicity, which is consistent with the toxicity of BSe nanoparticles[64]. Therefore, the choice of low and medium doses is crucial to ensure safety in vivo.

## Discussion

To meet the needs of different antibiofilm scenarios, the construction of safe and efficient antibacterial materials to combat drug-resistant biofilm-associated infections in a flexible manner is pressing. In particular, limited penetration greatly hinders the antibacterial effect due to the presence of shielding matrix of EPS in biofilms. Various approaches based on nanoparticles have been developed for the treatment of biofilm infections, including PTT. Generally speaking, nanoparticles are fabricated to eliminate biofilms through enhanced killing effect. Janus nanoparticles are attractive in antibacterial therapy since there could be more possibilities through the rational design of the two separate domains. For example, the near-field enhancement effect between porphyrin polymersome and gold in Janus Au-polymersome nanoparticles could be achieved to produce enhanced photothermal killing effect[37]. Janus nanomotor can also realize enhanced penetration in biofilm to kill the bacteria effectively[54]. On the other hand, on-demand biofilm dispersion offers an alternative avenue for eliminating biofilms, which remains to be explored. The strategy proposed in this work utilizes the specific interaction between EPS and dextran nanoparticles to fabricate nanoplatforms targeting biofilms. The rational design of Janus structure allows the maximum exposure of dextran domain while the introduction of BSe nanosheets can not only integrate photothermal property, but also achieve synergistic effects between the dextran and BSe components in Janus Dex-BSe nanoparticles. In addition, the positively charged Janus Dex-BSe nanoparticles are expected to realize biofilm removal by nanoscale bacterial debridement benefiting from the enhanced solubility of the bacterial-nanoparticle complex[24]. More interestingly, synergistic enhancement of gene regulation was found to facilitate biofilm dispersion mediated by Janus Dex-BSe. When NIR light was applied after the accumulation of Janus nanoparticles in biofilms, active motion induced by temperature gradient across the Janus boundary further enhance the photothermal effect of Janus Dex-BSe. As a result, biofilms could be efficiently eradicated by photothermal killing, which also indicates the synergistic enhancement of PTT. In a word, EPS targeting of Janus nanoparticles from the dextran domain benefits the interaction between Janus nanoparticles and biofilms through deep penetration while the inorganic BSe part imparts more possibility for the flexible elimination of biofilms with enhanced effectiveness. More importantly, both biofilm removal through dispersion and photothermal killing mediated by Janus Dex-BSe could be applied for the elimination of drug-resistant biofilms. As a proof of concept, Janus Dex-BSe nanoparticles were used in antibacterial wound dressing and the treatment of abscess in vivo, respectively.

In summary, Janus-structured Dex-BSe nanoparticles composed of biocompatible dextran and photothermal BSe nanosheets with synergistic effects were successfully prepared by a facile strategy. Thanks to the dextran domain, EPS targeting and penetration were realized by the Janus nanoparticles. Efficient accumulation and penetration of Janus Dex-BSe nanoparticles in the *S. aureus* biofilm were

observed in 30 min incubation. Then the biofilm was efficiently dispersed over time. In contrast, the CS-BSe counterparts did not induce any dispersal of the biofilm, validating the contribution of dextran shell to incorporate into the biofilm matrix and facilitate the bacterial detachment. More interestingly, RNA-seq transcriptomics revealed that genes related to extracellular proteases, amino acids synthesis and metabolism were significantly upregulated mediated by Janus Dex-BSe compared with Dex nanoparticles, suggesting the synergistic enhancement of gene regulation in the dispersal of biofilms. In addition, taking advantages of the unique Janus structure and photothermal property of BSe, enhanced photothermal effect of Janus Dex-BSe under NIR light irradiation resulted in photothermal elimination of *S. aureus* biofilms. More importantly, the remarkable dispersion and photothermal eradication of MRSA biofilms make Janus Dex-BSe nanoparticles promising against drug-resistant biofilm-associated infections. Furthermore, the antibiofilm performance of Janus Dex-BSe nanoparticles employing MRSA-infected mouse excisional wound and abscess models confirmed their feasibility for the dispersal or NIR-triggered eradication of biofilms in vivo, respectively. In addition, the favorable biosafety and biocompatibility of Janus Dex-BSe nanoparticles at the dose of 10 mg/kg guarantee their great potential against biofilm-associated infections in vivo. On the other hand, there are still challenges for the current proposed nanoparticles. Further studies are needed to assess the long-term toxicity caused by the accumulation of nanoparticles in organs. In addition, the specific interactions between Janus Dex-BSe and biological system and the underlying mechanism remain to be investigated. The current work provides a promising avenue for the fabrication of safe and efficient nanoparticles to flexibly treat drug-resistant biofilm-associated infections in different scenarios.

## Methods

### Materials
Selenium (99.99%), bismuth trichloride (99%), dextran (Mw 20000), and N,N'-Carbonyldiimidazole (CDI, 98%) were obtained from Energy Chemical Co., Ltd (Shanghai, China). Hydrochloric acid (HCl, 37 wt%), ethylene glycol, dimethyl sulfoxide (DMSO, >90%), ethanol (>99.7%), and ethylene diamine tetraacetic acid disodium salt (EDTA, 99%) were provided by Beijing Chemical Co., Ltd (Beijing, China). Glutaraldehyde (GA, 25 wt%) was obtained from Sinopharm Group Co. Ltd (China). Chitosan oligosaccharide (Mw ~5000) was obtained from Jinan Haidebei Marine Biological Engineering Co., Ltd. (Jinan, China). Polyvinyl pyrrolidone (PVP, Mw ~10000), ethylenediamine (98.0%), 3-(4,5-Dimethylthiazol-2yl)−2,5-diphenyl tetrazolium bromide (MTT) were purchased from Sigma-Aldrich Chemical Co. (St. Louis, MO, USA). Sulfo-Cyanine5.5 N-hydroxysuccinimide ester (Cy5.5-NHS) was obtained from Lumiprobe GmbH (Hannover, Germany). The Live/Dead BacLight bacterial viability kit (L7012) was bought from Invitrogen (LifeTechnologies, USA). The strains of *Escherichia coli* (ATCC 25922) and *Staphylococcus aureus* (CMCC (B) 26003) were obtained from Promega (Madison, USA). The methicillin-resistant *Staphylococcus aureus* (MRSA) was obtained from Chinese-Japan friendship hospital (Beijing, China). The lysogeny broth (LB) medium in all bacterial cultures was pre-treated by autoclaving at 120 °C for 20 min before use. L929 (ATCC CCL-1) cells were provided by American Type Culture Collection (ATCC). Milli-Q Ultrapure water (18.2 MΩ) was used in all experiments unless otherwise stated.

### Synthesis of Janus Dex-BSe and CS-BSe nanoparticles
For the synthesis of Dex-BSe nanoparticles, 3 mg of Dex-NH$_2$ was dissolved into 9.8 mL of deionized water, and then 2.25 mg of EDTA and 0.5 mg of BSe were added into the solution under magnetic stirring at room temperature. Finally, 6 mL of ethanol and 250 μL of glutaraldehyde (GA) solution (25 wt%) were added dropwise under vigorous stirring. After 4 h, the resultant Dex-BSe nanoparticles were collected by centrifugation (2655 × $g$, 10 min). Dex nanoparticles were synthesized following similar procedures except that 0.5 mg of BSe was replaced by 0.5 mg of water.

For the preparation of CS-BSe, 3 mg of chitosan was dissolved into 10.4 mL of deionized water, and then 12 mg of EDTA and 0.5 mg of BSe were added into the solution under magnetic stirring at room temperature. Finally, 7.8 mL of ethanol and 300 μL of GA solution were added dropwise under vigorous stirring. After 4 h, the resultant CS-BSe nanoparticles were collected by centrifugation (2655 × $g$, 10 min).

For Cy5.5 labeling, 10 μL of Cy5.5-NHS solution (5 mg/mL) was added to 1 mL of Dex-BSe or CS-BSe aqueous solution (5 mg/mL) and stirred for 12 h in dark. Thereafter, Cy5.5-labeled Dex-BSe and CS-BSe were collected by centrifugation (2655 × $g$, 10 min) and washed with ethanol until the supernatant was colorless.

### Motion analysis of Dex-BSe
The Dex-BSe solution in PBS (100 μg/mL) was added into a groove (5 mm of diameter, 0.5 mm of thickness) on the quartz slide (2 mm of thickness) sealed with a high-clean cover glass (8 mm of diameter). 808 nm NIR laser (1.0 W/cm$^2$) was employed to supply external light to trigger the motion of Dex-BSe. All the imaging experiments were recorded at a rate of 10 frames per second using a Nikon ECLIPSE Ni-U upright microscope, which was equipped with a 100W halogen tungsten lamp, a 20×plan fluor objective, and a CCD camera (DSRi2, Nikon). ImageJ was used to analyze the motion of nanomotors.

### Penetration of nanoparticles into biofilms
2 mL of *S. aureus* solution (1 × 10$^8$ CFU mL$^{-1}$) was incubated in LB medium at 37 °C without shaking for three days to allow the formation of mature biofilms. The bacterial suspensions were removed and the plates were washed with PBS for three times. To evaluate the penetration capacity of nanoparticles into biofilms, 1 mL of CS-BSe and Dex-BSe nanoparticles (512 μg/mL) labeled by Cy5.5 were added to *S. aureus* biofilms, respectively. The biofilms were stained by SYTO 9 (30 μL, 6 μmol/L) for 20 min in dark. The penetration of nanoparticles into *S. aureus* biofilms after 10, 30, and 60 min were observed using CLSM. To evaluate the penetration of nanoparticles under NIR light irradiation, 1 mL of Dex (307 μg/mL) and Dex-BSe (512 μg/mL) were added to *S. aureus* biofilms under NIR light irradiation (808 nm, 1.0 W/cm$^2$). The penetration of nanoparticles into *S. aureus* biofilms after exposure to NIR laser for 1, 3, and 5 min were observed using 3D confocal laser scanning microscope (CLSM, Leica, SP8).

### Dispersal of *S. aureus* biofilms
Biofilm dispersion was evaluated by CLSM, SEM, standard plate counting assays and crystal violet assay. To investigate biofilm dispersal effect, mature *S. aureus* or MRSA biofilms were cultured with Dex-BSe and CS-BSe at the BSe concentration of 205 μg/mL for 2, 4, and 6 h at 37 °C, respectively. Thereafter, the medium was removed and the residual *S. aureus* or MRSA biofilm was washed with PBS, stained by 30 μL of SYTO 9 for 20 min in dark, and then observed by CLSM. Biofilms treated with different nanoparticles for 6 h were also characterized by SEM.

To quantify the viable bacteria, residual biofilms treated with BSe (205 μg/mL), Dex (307 μg/mL), Dex+BSe (512 μg/mL), Dex-BSe (512 μg/mL), and CS-BSe (512 μg/mL) nanoparticles were sonicated and collected after 2, 4, and 6 h of incubation. The bacterial suspensions were serially diluted with sterile PBS. Then the dilution was plated on agar and incubated at 37 °C for 12 h and the residual bacteria in the biofilm were calculated.

To quantify the biomass, residual biofilms treated with different nanoparticles for 2, 4, and 6 h were stained by crystal violet. In brief, 1 mL of crystal violet (0.1%) was added into the biofilms and incubated at room temperature for 30 min. Then the medium was removed and the biofilms were washed with PBS. 500 μL of ethanol was

subsequently added and the optical density at 590 nm was measured. In addition, *S. aureus* biofilms after 2, 4, and 6 h of treatment were dispersed in PBS, centrifuged at 1699 × *g* for 3 min, dried overnight, and weighed.

For WGA staining, mature *S. aureus* biofilms were treated with different nanoparticles for 2, 4, and 6 h, respectively. The biofilms were then washed and stained with 200 μL of 5 μg/mL WGA-AF488 for 2 h at 4 °C in the darkness. After the unbound dye was removed, the bound dye was solved by 200 μL of 33% acetic acid. Then fluorescence at $\lambda_{excitation}$ = 495 nm and $\lambda_{emission}$ = 520 nm was measured with Bio-Rad Model 680 Microplate Reader after 1 h of incubation at 37 °C.

For EPS imaging, mature biofilms were incubated with WGA-AF488 for 2 h at 4 °C in darkness, and then 1 mL of Cy5.5-labeled nanoparticles were added, respectively. After 1 h of incubation, the unbound nanoparticles were removed and the biofilms were observed by CLSM.

For RNA-seq analysis, mature *S. aureus* biofilms were co-cultured with PBS, BSe (205 μg/mL), Dex (307 μg/mL), and Dex-BSe (512 μg/mL) at 37 °C for 6 h. Biofilms after different treatments were then collected for RNA-seq analysis by Beijing Genomics Institute (BGI) Co., Ltd on the BGISEQ-500 platform. Genes with FDR < 0.05 and |log₂ Fold change | ≥0.5 were considered as DEGs.

RT-PCR was performed on a BIOER Line Gene 9600 Plus (Bioer Technology) using mRNA/lncRNA qPCR Kit (GenePool), and the thermal cycling parameters were as follows: initial denaturation at 95 °C for 30 s, followed by 45 cycles at 95 °C for 5 s, 60 °C for 30 s, and 72 °C for 40 s. Each sample was analyzed in three duplicates. Data were analyzed using the ΔΔCt method, with 16 S rRNA as the internal reference gene. The primers used for RT-PCR were synthesized by Genepool Biotechnology (Beijing) Co., Ltd. and are listed in Supplementary Table 1.

### Bacterial killing and antibiofilm properties of Dex-BSe nanoparticles

To assess the antibacterial effect of Dex-BSe, 70 μL of bacterial solution ($2 \times 10^5$ CFU/mL) was treated with BSe, CS-BSe, and Dex-BSe nanoparticles at the BSe concentration of 410, 205, 102, 51, and 26 μg/mL, respectively. For NIR irradiation groups, the solution was exposed to an 808 nm laser at a power density of 1.0 W/cm² for 5 min. The bacterial solution was diluted for 10 times and 50 μL of the diluted bacterial solution was placed on LB agar and subjected to culturing at 37 °C. After 24 h, the bacterial colonies on the plate were analyzed. The antibacterial effect of BSe, CS-BSe and Dex-BSe against *E. coli* was also investigated following the similar procedures.

To measure the zeta potential of *S. aureus*, 20 μL of *S. aureus* ($10^8$ CFU/mL) dispersions were incubated with 20 μL of nanoparticle dispersions (1024 μg/mL) for 10 min. The zeta potential was examined by a Malvern Zetasizer Nano ZS90 instrument. To investigate the nanoparticle adhesion on bacterial surface by SEM imaging, after co-incubation with nanoparticles for 10 min, the bacteria were fixed with GA (2.5%) overnight. The fixed samples were serially dehydrated in gradient ethanol (25, 50, 75, 87.5 and 100%). Finally, the samples were dried and coated with platinum for SEM observation.

To evaluate the antibiofilm effect, fluorescence-based live/dead observation of biofilms was performed using SYTO 9 (6 μmol/L) and propidium iodide (PI, 30 μmol/L). After treatments with BSe (205 μg/mL), Dex-BSe (512 μg/mL) and CS-BSe (512 μg/mL) for 10 min, respectively, the biofilms were irradiation with NIR laser (808 nm, 1.0 W/cm², 5 min), and then stained for 20 min. SYTO 9 and PI were excited at 488 nm. The stained bacteria were imaged by 3D CLSM (Leica, SP8).

### Dispersal of MRSA biofilm by Dex-BSe in vivo

Female BALB/c mice (6–8 weeks, 16–20 g) were selected in the mouse excision wound model. Mice were housed in a temperature-constant animal room (22 °C) with a 12 h light/dark cycle and 30–70% humidity.

All animal experiments were approved by Ethical Committee of Chinese Academy of Medical Sciences and Peking Union Medical College. The mice were anesthetized using isoflurane and the round excision wounds with a diameter of 5 mm were created on the dorsal area. 10 μL of MRSA in PBS suspension ($10^8$ CFU/mL) was placed on wound site, which was covered by Tegaderm (3M™) to prevent contamination. For the preparation of hydrogels, 300 μL of 3 wt% sodium alginate aqueous solution was mixed with 100 μL of Dex-BSe, CS-BSe, and BSe aqueous solutions with BSe concentration of 2 mg/mL, respectively. Then 100 μL of 3 wt% calcium gluconate aqueous solution was added to the mixture. After 10 min, hydrogels containing Dex-BSe, CS-BSe, and BSe were formed. After biofilms were cultured for 24 h, Tegaderm film was removed and the first treatment was applied by covering the wound site with hydrogels containing nanoparticles. A new layer of Tegaderm was applied to immobilize the hydrogel and prevent contamination. Sodium alginate hydrogel was applied as negative control. After 6 h, the second and third treatments were subsequently applied with 6 h intervals. Finally, mice were sacrificed after the last treatment. Tissue samples were then harvested, homogenized, and plated on agar plates to determine bacterial counts.

### Photothermal antibiofilm effect in a MRSA-infected abscess model

Female BALB/c mice (6–8 weeks, 16–20 g) were employed in the abscess model. All animal experiments were approved by Ethical Committee of Chinese Academy of Medical Sciences and Peking Union Medical College. To evaluate antibiofilm effect of nanoparticles, MRSA-infected mice were randomly divided into three groups (*n* = 6) and intravenously injected with 100 μL of PBS, Dex-BSe, and CS-BSe (10 mg/kg), respectively. At 8 h post-injection, 808 nm NIR laser irradiation (1.0 W/cm², 5 min) was performed for the Dex-BSe and CS-BSe groups. The temperature was recorded by an infrared camera during the process of laser irradiation. All the mice were sacrificed after 7 days, and the infected tissues were collected for evaluation by the plate counting method. Skin tissues at the infected sites were harvested for histology analysis by H&E staining. On day 7, blood samples from the Dex-BSe group were collected for blood-index measurement. Finally, major organs including heart, liver, spleen, and kidney were collected for H&E staining.

### Safety assessment in vivo

The mice treated with PBS and different doses of Dex-BSe (3 mg/kg, 10 mg/kg and 50 mg/kg) were observed for 7 days. During the period, clinical observation including body weight was performed. After 7 days, the hippocampus and cerebral cortex of mice were collected for H&E staining and GSH, SOD activity and MDA content evaluation according to the kit protocols (Solarbio, China).

The mice treated with different doses of Dex-BSe after 7 days were anesthetized for the collection of BALF. The trachea was exposed by midline incision in the neck region. 1 ml of PBS was instilled into the lungs through trachea and withdrawn after 10 s. The recovered fluid was centrifuged at 4 × *g* for 5 min at 4 °C and the supernatant was utilized for the measurement of TNF-α, IL-6 and IL-1β by using ELISA kits. In addition, lung tissues after different treatments were collected and fixed in 10% formaldehyde for H&E staining. The serum after different treatments was collected to evaluate the renal function, and hepatic function.

### Statistical analysis

The experiment data were presented as means±standard deviation, where they were repeated at least three times. The differences between two groups were calculated by using two-tailed Student's *t*-test or one-way ANOVA using the Tukey post-test. In all tests, the exact *P* value is provided in the corresponding figure, and *P* values less than 0.05 was considered statistically significant. The statistical

analysis were performed using Microsoft Excle and GraphPad Prism software.

## Reporting summary

Further information on research design is available in the Nature Portfolio Reporting Summary linked to this article.

## Data availability

The RNA-Seq data generated in this study have been deposited in the NCBI Gene Expression Omnibus database under the accession code GSE239411. All other data supporting the findings of this study are available within the Article, Supplementary Information, or Source Date file. Source data are provided with this paper.

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

## Acknowledgements

This work was supported by the National Natural Science Foundation of China (Nos. 52173271 to N.Z., 52293382 and 52221006 to F.J.X.), Beijing Outstanding Young Scientist Program (No. BJJWZYJH01201910010024 to F.J.X.), Beijing Municipal Science and Technology Project (No. Z191100006619099 to N.Z.), and the Fundamental Research Funds for the Central Universities (No. BHYC1705A to N.Z.).

## Author contributions

Z.L., K.G., N.Z., and F.J.X. conceived and designed the concept of the experiments. Z.L., L.Y., and Y.W. synthesized the materials. Z.L., K.G., L.Y., K.Z., and Y.W. conducted the material characterizations. Z.L. and K.G. conducted and analyzed most of the in vitro and in vivo experi-ments. Z.L., K.G., and X.D. performed data curation. Z.L., K.G., N.Z., and F.J.X. wrote the paper. N.Z. and F.J.X. supervised the work. All authors discussed, commented, and agreed on the manuscript.

## Competing interests

The authors declare no competing interests.
