## [Peer Review File · Nature Communications]

Janus nanoparticles targeting extracellular polymeric substance achieve flexible elimination of drug-resistant biofilmsReviewers' Comments:

Reviewer #1:

Remarks to the Author:

In the manuscript entitled "Janus nanoparticles targeting extracellular polymeric substance achieve flexible elimination of drug-resistant biofilms", the authors are reporting asymmetrical Janus-structured dextran-bismuth selenide (Dex-BS) nanoparticles for accumulation at sites infected by pathogenic biofilms. The synthesized Janus nanocomposites are novel material, but the bismuth selenide nanoparticles and their potential cytotoxicity are already known in the references. Therefore, my attention and questions are focussed mainly to *in vivo* experiments i.e., the selectivity targeting, toxicity, retention in the blood stream and the photothermal performance of the Janus nanoparticles in mice experiments.

1. It is known that the bismuth selenide cause damage to the central nervous system, the liver and the digestive system. In addition, the nanoparticles caused also an immune-toxic effect over lungs and other secondary organs. Have the authors investigated the effect of the size and longevity in the blood stream on the nanoparticles distribution, accumulation in the body, as well as the efficacy of therapy in respect to the applied NIR energy dose? The dependence between dose-toxicity, or any further posttreatment side effects caused by accumulation, metabolism and bacterial destruction?
2. Is there any cytotoxicity effect to the normal cells or tissues due to generation of reactive oxygen species (after NIR irradiation), as well as any promoted oxidative stress?
3. The authors investigated the factors for targeting and accumulation of the Janus nanoparticles into the drug-resistant biofilms in the case of *in vitro* experiment. However, in the blood stream numerous globular proteins, blood cells and other bioagents might form stable complexes with the surface charges nanoparticles and thus inactivate them or remove from the body. What is the reason for targeting and accumulation of Janus nanoparticles in the infected sites but not in the other organs or healthy tissues?
4. How the nanoparticles and the dispersed biofilms are removed from the body after treatment?
2. Is there any cytotoxicity effect to the normal cells or tissues due to generation of reactive oxygen species (after NIR irradiation), as well as any promoted oxidative stress?
3. The authors investigated the factors for targeting and accumulation of the Janus nanoparticles into the drug-resistant biofilms in the case of *in vitro* experiment. However, in the blood stream numerous globular proteins, blood cells and other bioagents might form stable complexes with the surface charges nanoparticles and thus inactivate them or remove from the body. What is the reason for targeting and accumulation of Janus nanoparticles in the infected sites but not in the other organs or healthy tissues?
4. How the nanoparticles and the dispersed biofilms are removed from the body after treatment?

Reviewer #2:

Remarks to the Author:

The manuscript by Liu et al reports an alternative antibiofilm approach using asymmetrical Janus dextran-bismuth selenide (Dex-BS) nanoparticles that induce biofilm dispersal or NIR light-activated photothermal biofilm killing. The concept is interesting whereby the 'dextran domain' would enhance penetration and induce dispersion, while 'BS nanosheet domain' could promote active motion and photothermal killing effect via NIR light activation.

The synthesis of the Dex-BS nanoparticles appears to be straightforward using a previously published method, but characterization as well as data analyses are missing and need further clarifications (see additional comments). *In vitro* experiments demonstrate Dex-BS biofilm penetration and antibiofilm activity against *S. aureus* biofilms, including reduction of bacterial viability and biomass, although biofilm eradication is not achieved even after 6 h incubation.

Confocal imaging show reduction of EPS, while RNA-seq data and GO/KEGG analysis reveal putative genes/pathways affected that could explain the dispersion effects caused by the treatment. However, the transcriptomic analysis is based on a single time-point (end-point) while no experiments are done to validate the RNA-seq findings; simple assays could determine whether Dex-BS induces extracellular proteases and arginine metabolism as stated.

The antibiofilm performance is enhanced with NIR light irradiation, presumably due to BS nanosheet activation, achieving eradication. However, there is contradiction between viability data and confocal imaging data. For example, in Fig 5, viability test/colony counts show clear killing by both CS-BS (experimental control) and Dex-BS but Syto9/PI-based imaging shows that only Dex-BS is effective while CS-BS has minimal killing activity.

In vivo data from wound and abscess infection models appear to support therapeutic effects of Dex-BS but further clarifications and additional information are needed to evaluate interpretation and rigor of the biological data (see additional comments). There is a general lack of information on experimental conditions while the number of animals seems low to ensure adequate statistical analyses.

Overall, the study shows a very interesting multifunctional nanosystem with biofilm dispersion and biofilm killing properties via dextran- and photothermal-induced effects, but also needs more clarity about the experimental conditions/methodologies and data analysis to facilitate assessment and interpretation.

Additional comments:

1. Additional information about the Dex-BS nanostructures is needed, e.g. how uniform and reproducible are these Janus-like nanostructures? Does it always have one dextran domain and one BS nanosheet domain as depicted in Fig 1?
2. The Dex-BS shows Janus-like structure and photothermal properties but there is no data indicating active motion or nanomotor-like effect as stated.
3. The positive surface charge can inhibit biofilm penetration leading to accumulation on the biofilm outer-layers due to negatively charged bacteria. Any additional explanation on how it can achieve deeper penetration?
4. Also, it appears that the nanoparticles do not diffuse uniformly but rather start as initial patches of accumulation at specific locations on the surface and then diffuse across the biofilm over time. Any explanation for such spatiotemporal diffusion?
5. Although it is stated that photothermal effects do not compromise Dex-BS nanostructures, there is no data or images showing intact Dex-BS following photothermal activation.
6. Antibacterial assays and data analysis are inconsistent, confusing, and not well justified. Some figures show cfu data or just image of plate with colonies while in others % of bacterial viability or relative bacterial number is shown, making data interpretation difficult. Also, how the viability data is normalized?
7. The in vivo data on biofilm dispersal needs clarification. Based on the photographs, the initial wound in the Dex-BS group appears much milder at t0h (compared to CB-BS and control). Also, the wound progressively worsens over time. It is unclear how the relative bacterial number is determined and normalized. What is the number of viable bacteria at the wound site at t0h in each of the experimental groups?
8. The in vivo photothermal killing data is more convincing showing both significant bacterial killing as well as complementary analyses further supporting the antibiofilm effects including histopathology and blood biochemical indexes. What is the number of viable bacteria in the abscess prior to treatment? Is it similar across the different experimental groups?
9. The in vivo studies appear to be under powered with only 3 or 4 animals.
10. The discussion is limited. How this approach compares to other nano therapies as well as similarities and unique findings. Also, it would be nice to include discussions about limitations and how to address them.
11. There are some typographical as well as grammatical issues that require revision.

Reviewer #3:

Remarks to the Author:

In this work, Janus-structured nanoparticles targeting extracellular polymeric substances were proposed to achieve dispersion or NIR light-activated photothermal elimination of drug-resistant biofilms. It is impressive that the study demonstrates flexible applications of Janus Dex-BS nanoparticles in the MRSA-infected mouse excisional wound model and abscess model, respectively. The conclusions are well supported by the experimental results. Therefore, I recommend the publication of the manuscript in *Nature Communications* after the following questions are addressed.

1. The authors mentioned that the self-propelled active motion induced by the unique Janus structure was envisioned to enhance penetration depth. This should be verified by corresponding experiments. For example, the biofilm penetration by Dex and Dex-BS nanoparticles in the absence or presence of NIR light irradiation should be investigated. In addition, the photothermal killing of Dex-BS after different incubation times should be studied to verify the effect of penetration on killing effect.
2. Only Gram-positive bacteria *S. aureus* and MRSA were employed to demonstrate biofilm dispersion by Dex-BS. It would be beneficial to explore the dispersion effect on biofilms of Gram-negative bacteria.
3. For RNA-seq transcriptomics, it was found that the difference in gene expression identified in the Dex-BS group was more pronounced than in the Dex group. To further understand the underlying mechanism, DEGs between BS and the control group should also be analyzed to provide information whether BS has effect on biofilm-related genes.
4. In Fig 5c, the authors evaluated the photothermal killing effect of Dex-BS using CLSM. It is also necessary to perform quantitative analysis, such as CFU counts and biofilm biomass.
5. To verify the synergistic effect of biofilm dispersion and photothermal killing mediated by Dex-BS, the combination index or degree of synergy should be calculated.
6. Some citation numbers of the references are not superscripted, such as Nos. 19, 22, and 58.
7. It would be better to use “Dex-BSe” as the abbreviation of dextran-bismuth selenide.
8. In Fig. 7a, “removel” should be “removal”.

Response to Reviewers

Reviewer #1 (Remarks to the Author):

In the manuscript entitled "Janus nanoparticles targeting extracellular polymeric substance achieve flexible elimination of drug-resistant biofilms ", the authors are reporting asymmetrical Janus-structured dextran-bismuth selenide (Dex-BS) nanoparticles for accumulation at sites infected by pathogenic biofilms. The synthesized Janus nanocomposites are novel material, but the bismuth selenide nanoparticles and their potential cytotoxicity are already known in the references. Therefore, my attention and questions are focused mainly to in vivo experiments i.e., the selectivity targeting, toxicity, retention in the blood stream and the photothermal performance of the Janus nanoparticles in mice experiments.

Response: We appreciate the comments and valuable suggestions from the reviewer, which definitely help us improve the quality of our work.

1. It is known that the bismuth selenide cause damage to the central nervous system, the liver and the digestive system. In addition, the nanoparticles caused also an immune-toxic effect over lungs and other secondary organs. Have the authors investigated the effect of the size and longevity in the blood stream on the nanoparticles distribution, accumulation in the body, as well as the efficacy of therapy in respect to the applied NIR energy dose? The dependence between dose-toxicity, or any further posttreatment side effects caused by accumulation, metabolism and bacterial destruction?

Response: We are grateful to the reviewer for the comment. We agree with the reviewer that the particle size is crucial for the nanoparticles distribution and accumulation in the body (Refs: *Nat. Biotechnol.* 33, 941–951 (2015), *Biomaterials* 31, 3657–3666 (2010), et al.). In order to investigate the size effect, in addition to Janus Dex-BSe nanoparticles with the particle size of 200 nm, Janus Dex-BSe nanoparticles with a smaller size of 100 nm (Dex-BSe₁₀₀, Fig. R1a) and a larger size of 300 nm (Dex-BSe₃₀₀, Fig. R1b) were synthesized by varying the amount of ethylenediaminetetraacetic acid. As exhibited in Fig. R1c, the increase in the size of Dex-BSe₁₀₀, Dex-BSe, and Dex-BSe₃₀₀ nanoparticles was confirmed by dynamic light scattering.

Fig. R1. TEM images of (a) Dex-BSe₁₀₀ and (b) Dex-BSe₃₀₀ nanoparticles. c Size distribution of Dex-BSe₁₀₀ Dex-BSe, and Dex-BSe₃₀₀ determined by dynamic light scattering.

As shown in Fig. R2a, a time-dependent accumulation behavior at the infected sites was observed after intravenous injection of Cy5.5-labeled Dex-BSe₁₀₀, Dex-BSe, and Dex-BSe₃₀₀ nanoparticles, while the maximal accumulation appeared at 8 h post-injection. Further fluorescence images and quantitative analysis of ex vivo tissues demonstrated the accumulation of nanoparticles with different sizes in the infected abscesses was comparable at 8 h post-injection (Fig. R2b and R2c). In addition, the distribution of nanoparticles in the main organs including liver, spleen, and lung was comparable. Notably, Dex-BSe₁₀₀-treated group exhibited significantly stronger fluorescence in heart, kidney, and brain than that of the Dex-BSe- and Dex-BSe₃₀₀-treated groups, which reflects the size-dependent distribution of nanoparticles in the body. However, the high accumulation of Dex-BSe₁₀₀ in brain, heart, and kidney may lead to toxicity and potential risk (Refs: *Environ. Chem. Lett.* **18**, 1659–1683 (2020), *ACC. Chem. Res.* **52**, 1632–1642 (2019), et al.).

Fig. R2. **a** Time-dependent fluorescence images of MRSA biofilm-infected mice after the intraveous injection of Dex-BSe₁₀₀, Dex-BSe and Dex-BSe₃₀₀, respectively. Fluorescence images **(b)** and quantitative analysis **(c)** of major organs and the infected abscess after different treatments at 8 h post-injection. Data are presented as mean values \pm SD ($n = 5$ independent samples). Statistical significance was calculated by one-way ANOVA using the Tukey post-test.

To investigate the effect of the nanoparticles size on therapeutic efficacy, bacterial counts in infected abscesses after different treatments were evaluated by plate counting method. As shown in Fig. R3, negligible difference was found in the efficacy of photothermal antibiofilm therapy mediated by Dex-BSe nanoparticles with different sizes in vivo, which could be attributed to the comparable accumulation of Dex-BSe₁₀₀, Dex-BSe, and Dex-BSe₃₀₀ (Fig. R2c).

Fig. R3. Photographs (a) and quantitative analysis (b) of bacterial colonies in abscesses of different treatment groups after 7 days. Data are presented as mean values \pm SD ($n = 5$ independent samples).

The reviewer raised an important question about in vivo safety issue. The dose-dependent toxicity and potential side effects of the Janus Dex-BSe nanoparticles were then evaluated. No obvious difference in body weight was observed between the control group and Dex-BSe-treated groups with different doses (3, 10, and 50 mg/kg Dex-BSe, Supplementary Fig. 23), indicating Dex-BSe nanoparticles have no evident adverse effects on mice. Oxidative stress has been considered as a major concern for nanoparticle-induced neurotoxicity^{62,63}. To investigate the toxicity of different doses of Dex-BSe on the central nervous system of mice, the activity of superoxide dismutase (SOD), glutathione (GSH), and the malondialdehyde (MDA) level in the central nervous system were measured, which are closely correlated with oxidative stress. As shown in Supplementary Fig. 24, the GSH and SOD activity, as well as the MDA level of hippocampus and cerebral cortex in each dose group after 7 days of treatment did not show obvious changes compared with the control group, indicating no noticeable oxidative injuries was induced. Moreover, no obvious damage to neurons was observed in the hippocampus (CA1, CA3, and dentate gyrus (DG) regions) and cerebral cortex after different treatments (Supplementary Fig. 25), confirming compromised toxicity of Dex-BSe to the central nervous system.

Supplementary Fig. 23. Body weight changes of MRSA-infected mice after treatment with different doses of Dex-BSe nanoparticles. Data are presented as mean values \pm SD ($n = 5$ independent samples). Source data are provided as a Source Data file.

Supplementary Fig. 24. The activity of (a) glutathione (GSH) and (b) superoxide dismutase (SOD), and (c) the malondialdehyde (MDA) level in hippocampus treated with different doses of Dex-BSe after 7 days. The activity of (d) GSH and (e) SOD, and (f) the MDA level in cerebral cortex after treatment with different doses of Dex-BSe after 7 days. Data are presented as mean values \pm SD ($n = 5$ independent samples). Source data are provided as a Source Data file.

Supplementary Fig. 25. H&E staining images of hippocampus and cerebral cortex of mice treated with different doses of Dex-BSe after 7 days.

In order to evaluate the dose-dependent toxicity and potential effects on lung, bronchoalveolar lavage fluid (BALF) after the treatment of different doses of Dex-BSe was collected for the detection of pro-inflammatory cytokines, which could be induced by cellular damage via oxidative stress^{64,65}. As shown in Supplementary Fig. 26, compared with the control group, the levels of pro-inflammatory cytokines including tumor necrosis factor-alpha (TNF- α), interleukin-6 (IL-6), and interleukin-1 β (IL-1 β) were found to increase significantly in BALF of mice treated with high dose of Dex-BSe (50 mg/kg), while no obvious changes were observed in the low and medium dose groups (3 and 10 mg/kg of Dex-BSe), demonstrating a dose-dependent oxidative damage. Histological analysis of the lungs indicates negligible lung damage or fibrosis (Supplementary Fig. 27). However, pro-inflammatory cells infiltration was observed in the high dose group while negligible inflammation was found in the low and medium groups, verifying the pulmonary inflammation induced by high dose of Dex-BSe via oxidative stress. No significant inflammation-associated spleen enlargement was detected in the low and medium doses group, while obvious enlargement and a significant increase of spleen index were observed in the high dose treatment group (Supplementary Fig. 28), implying the possible spleen toxicity caused by high dose of Dex-BSe. Additionally, markers for hepatic (alanine aminotransferase (ALT) and aspartate aminotransferase (AST)) and renal function (creatinine (CREA) and urea nitrogen (UREA)) were studied to evaluate the toxicity of Dex-BSe. As displayed in Supplementary Fig. 29, no obvious changes in ALT, AST, CREA, and UREA were observed after 7 days of treatments compared with the control group, indicating the normal liver and kidney function of mice after the treatment with different doses of Dex-BSe. Collectively, Dex-BSe nanoparticles may not cause apparent damage to the central nervous system, liver or kidneys while high doses of Dex-BSe nanoparticles may induce potential pulmonary and splenic toxicity, which is consistent with the toxicity of BSe nanoparticles⁶⁴. Therefore, the choice of low and medium doses is crucial to ensure safety *in vivo*.

Relevant information has been added in **Lines 518-562** and **Supplementary Figs. 23-29**.

Supplementary Fig. 26. The levels of TNF- α (a), IL-6 (b) and IL-1 β (c) in bronchoalveolar lavage fluid after different treatments. Data are presented as mean values \pm SD ($n = 5$ independent samples). Statistical significance was calculated by one-way ANOVA using the Tukey post-test. Source data are provided as a Source Data file.

Supplementary Fig. 27. H&E staining images of lung tissues of mice in the control group (a) and treated with different doses of Dex-BSe (b) 3 mg/kg, (c) 10 mg/kg, and (d) 50 mg/kg.

Supplementary Fig. 28. a Photograph of spleens collected from mice after 7 days of treatment with different doses of Dex-BSe. b Quantification of the corresponding spleen/body weight. Data are presented as mean values \pm SD ($n = 5$ independent samples). Statistical significance was calculated by one-way ANOVA using the Tukey post-test. Source data are provided as a Source Data file.

Supplementary Fig. 29. Kidney and liver function markers of mice after 7 days of treatment with different doses of Dex-BSe. (ALT: alanine aminotransferase; AST: aspartate aminotransferase; UREA: urea nitrogen; CREA: creatinine). Data are presented as mean values \pm SD ($n = 5$ independent samples). Source data are provided as a Source Data file.

2. Is there any cytotoxicity effect to the normal cells or tissues due to generation of reactive oxygen species (after NIR irradiation), as well as any promoted oxidative stress?

Response: We appreciated the comment from the reviewer. Firstly, methylene blue (MB) was employed as the reactive oxygen species (ROS) indicator to verify the ROS products. As shown in Fig. R4, no obvious MB degradation was observed in the presence of Dex-BSe under NIR irradiation, suggesting negligible generation of ROS. In addition, to investigate the cytotoxicity effect to the normal cells after NIR irradiation, the performance of ROS generation in normal cells incubated with Dex-BSe was investigated by fluorescence microscope employing 2',7'-dichlorofluorescein diacetate (DCFH-DA) as a probe. Similarly, no noticeable green fluorescence of DCF (2',7'-dichlorofluorescein) in L929 cells was observed (Fig. R5), verifying the negligible generation of ROS. Meanwhile, compromised cytotoxicity effect to the normal cells was observed when the experiments were carried out at 4 °C to minimize the photothermal effect-induced cell apoptosis. Therefore, there is not any cytotoxicity effect to the normal cells or tissues due to the negligible generation of ROS after NIR irradiation.

Fig. R4. Degradation behaviors of MB in the presence of Dex-BSe with different concentrations with or without NIR light irradiation.

Fig. R5. Representative fluorescent and bright field images of DCFH-DA stained L929 cells in the presence or absence of Dex-BSe under NIR irradiation at 4 °C.

3. The authors investigated the factors for targeting and accumulation of the Janus nanoparticles into the drug-resistant biofilms in the case of in vitro experiment. However, in the blood stream numerous globular proteins, blood cells and other bioagents might form stable complexes with the surface charges nanoparticles and thus inactivate them or remove from the body. What is the reason for targeting and accumulation of Janus nanoparticles in the infected sites but not in the other organs or healthy tissues?

Response: We appreciate the comment from the reviewer. Janus Dex-BSe nanoparticles could penetrate deep layers of biofilms due to the characteristic of dextran domain incorporation into the biofilm matrix via bacterial exoenzymes, which is consistent with previous reports that dextran-functionalized nanoparticles can target and penetrate into biofilms (Refs: *Nano Today*. **38**, 101118 (2021) and *ACS Nano* **13**, 4960–4971 (2019)). We agree with the reviewer that the targeting and accumulation of Janus nanoparticles in vivo might be influenced by proteins, blood cells and other bioagents in the blood stream (Refs: *Nat. Nanotechnol.* **16**, 708–716 (2021), *Adv. Mater.* **31**, 1805740 (2019), *J. Am. Chem. Soc.* **142**, 8827–8836 (2020), *ACS Nano* **15**, 15397–15401 (2021), et al.). Actually, the distribution of nanoparticles in main organs and infected abscesses visualized at 8 h post-injection (**Fig. 8c**) indicates that in

addition to the high accumulation of Dex-BSe in the MRSA biofilm-infected abscess, considerable nanoparticle was also found in liver at this stage, which is consistent with previous reports that metabolizable BSe nanoparticles were mainly cleared by the liver (Refs: *Adv. Funct. Mater.* **24**, 1718–1729 (2014) and *Small* **12**, 4136–4145 (2016)). These results are in agreement with the accumulation of other nanoparticles in the infected sites and the other organs (Refs: *Nat. Biomed. Eng.* **2**, 95–103 (2018), *ACS Nano* **14**, 347–359 (2020), and *Nat. Commun.* **13**, 7164 (2022) et al.).

4. How the nanoparticles and the dispersed biofilms are removed from the body after treatment?

Response: We thank the reviewer for this question that reminds us to further explore the removal of the nanoparticles and the dispersed biofilms from the body after treatment. Firstly, the amount of Dex-BSe nanoparticles in the hydrogel dressing, feces, and urine were evaluated respectively by inductively coupled plasma-mass spectrometry (ICP-MS). As shown in Fig. R6a, approximately 70% of nanoparticles remained in the hydrogel dressings. Dex-BSe nanoparticles that entered the body could be excreted from both urine and feces, but mainly from feces (Fig. R6b), which is consistent with the excretion of BSe nanoparticles from the body (Refs: *Adv. Funct. Mater.* **24**, 1718–1729 (2014) and *Small* **12**, 4136–4145 (2016)).

The removal of the dispersed biofilms from the body was realized by the hydrogel dressings containing Dex-BSe nanoparticles covered on the wound (Fig. R6c). The number of bacteria in the hydrogel dressings containing Dex-BSe was significantly higher than that in the control groups at different time points (Fig. R6d), indicating efficient removal of the dispersed biofilms by hydrogel dressings containing nanoparticles. In addition, the small number of bacteria remaining in the wound can be cleared by immunity the host immune response (Refs: *Cell* **124**, 783–801 (2006), *Nat. Rev. Microbiol.* **18**, 571–586 (2020), *Nat. Nanotechnol.* **15**, 41–46 (2020), *Nat. Commun.* **12**, 6143 (2021), et al.)

Fig. R6. Quantitative analysis of the Bi content remained in hydrogel dressings (a), as well as in urine and feces (b) at different time points after treatment. c The bacteria removed by hydrogel dressings after different treatments were cultured on agar plates. d Bacterial colony counting after different treatments. Data are normalized by bacterial counts in the wound tissue without treatments. Data are presented as mean values \pm SD ($n = 6$ independent samples). Statistical significance was calculated by two-tailed Student's t -test.

From Reviewer #2 (Remarks to the Author):

The manuscript by Liu et al reports an alternative antibiofilm approach using asymmetrical Janus dextran-bismuth selenide (Dex-BS) nanoparticles that induce biofilm dispersal or NIR light-activated photothermal biofilm killing. The concept is interesting whereby the 'dextran domain' would enhance penetration and induce dispersion, while 'BS nanosheet domain' could promote active motion and photothermal killing effect via NIR light activation.

Response: We appreciate the positive comments and valuable suggestions from the reviewer, which definitely help us improve the quality of our work.

*The synthesis of the Dex-BS nanoparticles appears to be straightforward using a previously published method, but characterization as well as data analyses are missing and need further clarifications (see additional comments). In vitro experiments demonstrate Dex-BS biofilm penetration and antibiofilm activity against *S. aureus* biofilms, including reduction of bacterial viability and biomass, although biofilm eradication is not achieved even after 6 h incubation.*

Response: We are grateful to the reviewer for the comment. As suggested, characterization as well as data analyses as listed in additional comments have been supplemented in the revised manuscript. For the antibiofilm activity of Dex-BSe against *S. aureus* biofilms in vitro, ~85% of the biofilm was removed after 6 h of treatment (Fig. 3f), which is comparable with the previous reported biofilm eradication efficiency (Refs: *Nano Lett.*, **20**, 7350–7358 (2020), *Adv. Funct. Mater.* **29**, 1808222 (2019), et al.). With longer treatment times, the biofilm eradication could be improved due to the excellent dispersion capability of Dex-BSe. After 24 h, considerable reduction in *S. aureus* biofilm biomass could be achieved and more than 95% of the biofilm mass was removed (Fig. R7).

Fig. R7. Quantitative analysis of the crystal violet-stained *S. aureus* biofilms treated with Dex-BSe at different incubation times. Data are presented as mean values \pm SD ($n = 3$ independent samples). Statistical significance was calculated by two-tailed Student's *t*-test.

Confocal imaging show reduction of EPS, while RNA-seq data and GO/KEGG analysis reveal putative genes/pathways affected that could explain the dispersion effects caused by the treatment. However, the transcriptomic analysis is based on a single time-point (end-point) while no experiments are done to validate the RNA-seq findings; simple assays could determine whether Dex-BS induces extracellular proteases and arginine metabolism as stated.

Response: We thank the reviewer for the reminder. RT-PCR assay was further performed to validate the RNA-seq findings. The expressions levels of typical genes associated with extracellular proteases (*sspA* and *sspB*) and arginine biosynthesis and arginine catabolism (*arcD*, *arcC*, *argH*, and *argF*) were remarkably upregulated after Dex-BSe treatment, which exhibited a high consistency with RNA-seq results (Supplementary Fig. 13). Corresponding information has been added in **Lines 315-317** and **Supplementary Fig. 13**.

Supplementary Fig. 13. RT-PCR results of typical genes encoding extracellular proteases (a,b) and involved in arginine biosynthesis and arginine catabolism (c-f). Data are presented as mean values \pm SD ($n = 3$ independent samples). Statistical significance was calculated by one-way ANOVA using the Tukey post-test. Source data are provided as a Source Data file.

The antibiofilm performance is enhanced with NIR light irradiation, presumably due to BS nanosheet activation, achieving eradication. However, there is contradiction between viability data and confocal imaging data. For example, in Fig 5, viability test/colony counts show clear killing by both CS-BS (experimental control) and

Dex-BS but Syto9/PI-based imaging shows that only Dex-BS is effective while CS-BS has minimal killing activity.

Response: We are grateful to the reviewer. The viability test/colony counts in Fig. 5a,b show the antibacterial results of nanoparticles against planktonic *S. aureus*. Comparable antibacterial activity of Dex-BSe and CS-BSe at the same BSe concentration was found due to their comparable photothermal killing effect (Fig. 2g). On the other hand, SYTO 9/PI-based imaging in Fig. 5c reveals the antibiofilm performance of nanoparticles. Dex-BSe nanoparticles demonstrated effective antibiofilm effect due to their excellent biofilm penetration. In contrast, the minimal killing activity mediated by CS-BSe was supposed to be caused by the limited biofilm penetration, as shown in Fig. 3a.

In vivo data from wound and abscess infection models appear to support therapeutic effects of Dex-BS but further clarifications and additional information are needed to evaluate interpretation and rigor of the biological data (see additional comments). There is a general lack of information on experimental conditions while the number of animals seems low to ensure adequate statistical analyses.

Response: We thank the reviewer for the reminder. As suggested, detailed information on experimental conditions has been provided in the revised manuscript. In addition, the number of animals has been increased in our experiments to ensure adequate statistical analyses. Please see the response to additional comments.

Overall, the study shows a very interesting multifunctional nanosystem with biofilm dispersion and biofilm killing properties via dextran- and photothermal-induced effects, but also needs more clarity about the experimental conditions/methodologies and data analysis to facilitate assessment and interpretation.

Response: We appreciate the positive comments and valuable suggestions from the reviewer. We have made clarity about the experimental conditions/methodologies and data analysis to facilitate assessment and interpretation in the revised manuscript.

Additional comments:

1. Additional information about the Dex-BS nanostructures is needed, e.g. how uniform and reproducible are these Janus-like nanostructures? Does it always have one dextran domain and one BS nanosheet domain as depicted in Fig 1?

Response: We thank the reviewer for the reminder. As suggested, the size distribution calculated by the measurement of more than 200 nanoparticles shows that monodisperse Janus Dex-BSe nanoparticles with an average size of ~220 nm were prepared (Supplementary Fig. 2a). In addition, the Janus nanostructures are highly reproducible. Statistical results indicate that a high yield of 78.1% for Janus-structured Dex-BSe with a well-defined single BSe domain on a dextran nanosphere (Supplementary Fig. 2b). Related information has been supplemented in **Lines 147-151 and Supplementary Fig. 2.**

Supplementary Fig. 2 a Size distribution of Dex-BSe nanoparticles. b Statistical results of dextran domain loaded with different numbers of BSe domains.

2. The Dex-BS shows Janus-like structure and photothermal properties but there is no data indicating active motion or nanomotor-like effect as stated.

Response: We appreciate the comment from the reviewer. To verify the self-propelled active motion of Janus Dex-BSe nanomotors, the trajectories of Dex-BSe under NIR irradiation were recorded. As shown in Supplementary Fig. 6a and Supplementary Movie 1, Dex-BSe exhibited nondirectional Brownian motion in the absence of NIR irradiation. In contrast, the rapid propulsion of Dex-BSe was observed under NIR laser irradiation. Accordingly, the average velocity of the Dex-BSe increased significantly from 2.0 to 3.9 $\mu\text{m/s}$ after NIR irradiation was applied (Supplementary Fig. 6b). The NIR-triggered active motion induced by temperature gradient across the Janus boundary contributed to the enhanced photothermal effect of Dex-BSe in the medium containing *S. aureus*. The corresponding information has added in **Lines 187-195** and **Supplementary Fig. 6**, and **Supplementary Movie 1**.

Supplementary Fig. 6 a Trajectories of Dex-BSe in PBS in the absence and presence of NIR laser irradiation (1.0 W/cm^2 , 30 s). b The average velocity of Dex-BSe at different conditions. Data are presented as mean values \pm SD ($n = 3$ independent samples). Statistical significance was calculated by two-tailed Student's *t*-test. Source data are provided as a Source Data file.

3. The positive surface charge can inhibit biofilm penetration leading to accumulation on the biofilm outer-layers due to negatively charged bacteria. Any additional

explanation on how it can achieve deeper penetration?

Response: We thank the reviewer for the comment. We agree with the reviewer that nanoparticles with the positive surface charge are often unable to penetrate into the biofilm but rather accumulate on the surface of the biofilm (Ref: *Sci. Adv.* **6**, 1112, (2020)). On the other hand, the nanoparticles that interact with the bacteria or biofilm matrix likely penetrate into the biofilm by means of diffusion (Refs: *J. Control. Release* **190**, 607–623 (2014), *Ann. Biomed. Eng.* **41**, 53–67, (2013), et al.). In this work, Janus Dex-BSe nanoparticles could penetrate deep layers of biofilms due to the characteristic of dextran domain that interact with the biofilm matrix by exoenzymes for the synthesis of EPS glucans, which is consistent with previous reports that dextran-functionalized nanoparticles can target and penetrate into biofilms (Refs: *ACS Appl. Mater. Interfaces* **13**, 29269–29280 (2021), *ACS Nano* **13**, 4960–4971 (2019), et al.). In addition, the self-propelled active motion induced by the unique Janus structure of Dex-BSe under NIR irradiation is thought to realize enhanced penetration into biofilm, which is consistent with previous works (Refs: *Nano Lett.*, **20**, 7350–7358 (2020), *Small* **18**, 2205252 (2022)).

4. Also, it appears that the nanoparticles do not diffuse uniformly but rather start as initial patches of accumulation at specific locations on the surface and then diffuse across the biofilm over time. Any explanation for such spatiotemporal diffusion?

Response: We are grateful to the reviewer for the comment. As stated previously, Dex-BSe nanoparticles diffuse into the biofilm with enhanced penetration through the interaction with the biofilm matrix, which is physicochemically complex and extremely heterogeneous over small spatial scale. The feature of EPS matrix significantly impact nanoparticle-biofilm interactions while the complex composition in EPS influences the initial deposition and subsequent accumulation of nanoparticles. Once nanoparticles bind to EPS, they can then migrate deeper into the matrix by diffusion over time. In addition, the pores and crevices in the biofilm may deform to allow the nanoparticles to diffuse (Refs: *Trends Microbiol.* **27**, 11, (2019); *Front. Microbiol.* **6**, 591 (2015); *Ann. Biomed. Eng.* **41**, 53–67 (2013), et, al.). Therefore, the nanoparticles could not diffuse uniformly but rather start as initial patches of accumulation at specific locations on the surface and then diffuse across the biofilm over time.

5. Although it is stated that photothermal effects do not compromise Dex-BS nanostructures, there is no data or images showing intact Dex-BS following photothermal activation.

Response: We thank the reviewer for the reminder. As suggested, TEM was performed to observe the morphology of Dex-BSe following photothermal activation. In addition, the morphology and size of Dex-BSe after NIR irradiation did not change noticeably (Supplementary Fig. 5d,e), verifying the stability of Dex-BSe for promising photothermal agents. Related information has been added in **Lines 178-180** and **Supplementary Fig. 5d**.

Supplementary Fig. 5d TEM image of Dex-BSe after 808 nm laser irradiation (1 W/cm², 5 min).

6. Antibacterial assays and data analysis are inconsistent, confusing, and not well justified. Some figures show cfu data or just image of plate with colonies while in others % of bacterial viability or relative bacterial number is shown, making data interpretation difficult. Also, how the viability data is normalized?

Response: We thank the reviewer for the comment. As suggested, we utilized bacterial viability for in vitro test and relative bacterial number for in vivo test, respectively to facilitate data interpretation. The viability data are normalized to make such data easily comparable. To realize the normalization, the bacterial number of the PBS control is set to 100%, and all data in the treatment groups are indicated relative to this value.

7. The in vivo data on biofilm dispersal needs clarification. Based on the photographs, the initial wound in the Dex-BS group appears much milder at t0h (compared to CB-BS and control). Also, the wound progressively worsens over time. It is unclear how the relative bacterial number is determined and normalized. What is the number of viable bacteria at the wound site at t0h in each of the experimental groups?

Response: We are grateful to the reviewer for the comment. To clarify the in vivo data on biofilm dispersal, we performed further experiment with increased number of animals per group ($n = 6$ independent samples). To ensure the initial wounds with similar area in each group, the round excision wounds were created on the dorsal area using a 5 mm diameter biopsy punch. Representative time-dependent photographs of wounds are shown in Fig. 7c. Furthermore, the relative wound areas were quantified by the ImageJ software and the results of relative wound area were normalized to the wound area at 0 h. As shown in Fig. R8, no noticeable changes were found in different groups during the 18 h treatment period.

Fig. 7c Time-dependent photographs of wounds with different treatments.

Fig. R8. The relative wound area proportion of the wounds with different treatments at 6, 12, and 18 h. Data are presented as mean values \pm SD ($n = 6$ independent samples).

To determine the number of bacteria in the wound tissue, the wound tissues collected from each group were soaked in 1 mL of sterile PBS and homogenized. Then the bacterial suspensions were subjected to dilution and plated on LB agar plates. Following incubation at 37 °C for 24 h, the bacterial colonies were counted. To determine and normalize the relative bacterial number, the bacterial number of the control group was set to 100%, and all data in the treatment groups were indicated relative to this value.

The number of viable bacteria at the wound site at 0 h in each experimental group was quantified by spread plate method. A comparable bacterial number of $\sim 5 \times 10^8$ CFU were found in each group (Fig. R9).

Fig. R9. Numbers of bacteria at the wound sites before different treatments. Data are presented as mean values \pm SD ($n = 5$ independent samples). Statistical significance was calculated by one-way ANOVA using the Tukey post-test.

8. The in vivo photothermal killing data is more convincing showing both significant bacterial killing as well as complementary analyses further supporting the antibiofilm effects including histopathology and blood biochemical indexes. What is the number of viable bacteria in the abscess prior to treatment? Is it similar across the different experimental groups?

Response: We are grateful to the reviewer for the comment. The number of viable bacteria in the abscess prior treatment was determined to be $\sim 4.3 \times 10^9$ CFU by the plate counting method (Fig. R10). Statistical analysis shows that there was no significant difference across the different experimental groups.

Fig. R10. Numbers of bacteria in the abscesses before different treatments. Data are presented as mean values \pm SD ($n = 5$ independent samples). Statistical significance was calculated by one-way ANOVA using the Tukey post-test.

9. The in vivo studies appear to be under powered with only 3 or 4 animals.

Response: We thank the reviewer for the reminder. As suggested, we performed further in vivo experiments with increased number of animals per group ($n = 6$ independent samples). Corresponding in vivo data can be found in **Figs. 7,8**.

10. The discussion is limited. How this approach compares to other nano therapies as well as similarities and unique findings. Also, it would be nice to include discussions about limitations and how to address them.

Response: We appreciate the comment from the reviewer. As suggested, we have supplemented more discussion. Various approaches based on nanoparticles have been developed for the treatment of biofilm infections, including PTT. Generally speaking, nanoparticles are fabricated to eliminate biofilms through enhanced killing effect. Janus nanoparticles are attractive in antibacterial therapy since there could be more possibilities through the rational design of the two separate domains. For example, the near-field enhancement effect between porphyrin polymersome and gold in Janus Au-polymersome nanoparticles could be achieved to produce enhanced photothermal killing effect.³⁷ Janus nanomotor can also realize enhanced penetration in biofilm to kill the bacteria effectively.⁵⁴ On the other hand, on-demand biofilm dispersion offers a new avenue for eliminating biofilms, which remains to be explored. The strategy proposed in this work utilizes the specific interaction between EPS and dextran nanoparticles to fabricate nanoplatforms targeting biofilms. The rational design of Janus structure allows the maximum exposure of dextran domain while the introduction of BSe nanosheets can not only integrate photothermal property, but also achieve synergistic effects between the dextran and BSe components in Janus Dex-BSe nanoparticles. In addition, the positively charged Janus Dex-BSe nanoparticles are expected to realize biofilm removal by nanoscale bacterial debridement.

More importantly, both biofilm removal through dispersion and photothermal killing mediated by Janus Dex-BSe could be applied for the elimination of drug-resistant biofilms. As a proof of concept, Janus Dex-BSe nanoparticles were used in antibacterial wound dressing and the treatment of abscess in vivo, respectively.

On the other hand, there are still challenges for the current proposed nanoparticles. Further studies are needed to assess the long-term toxicity caused by the accumulation of nanoparticles in organs. In addition, the specific interactions between Janus Dex-BSe and biological system and the underlying mechanism remain to be investigated. The current work provides a new avenue for the fabrication of safe and efficient nanoparticles to flexibly treat drug-resistant biofilm-associated infections in different scenarios.

Corresponding information has been supplemented in **Lines 568-586, 595-599 and 622-628**.

11. There are some typographical as well as grammatical issues that require revision.

Response: We are grateful to the reviewer and apologize for the typographical and grammatical issues. All of these issues have been carefully checked to improve the quality of the manuscript.

From Reviewer #3 (Remarks to the Author):

In this work, Janus-structured nanoparticles targeting extracellular polymeric

substances were proposed to achieve dispersion or NIR light-activated photothermal elimination of drug-resistant biofilms. It is impressive that the study demonstrates flexible applications of Janus Dex-BS nanoparticles in the MRSA-infected mouse excisional wound model and abscess model, respectively. The conclusions are well supported by the experimental results. Therefore, I recommend the publication of the manuscript in *Nature Communications* after the following questions are addressed.

Response: We thank the reviewer for the positive comments and valuable suggestions, which definitely help us improve the quality of our work.

1. The authors mentioned that the self-propelled active motion induced by the unique Janus structure was envisioned to enhance penetration depth. This should be verified by corresponding experiments. For example, the biofilm penetration by Dex and DexBS nanoparticles in the absence or presence of NIR light irradiation should be investigated. In addition, the photothermal killing of Dex-BS after different incubation times should be studied to verify the effect of penetration on killing effect.

Response: We appreciated the comment from the reviewer. As suggested, to verify enhanced biofilm penetration induced by the self-propelled active motion, the biofilm penetration by Dex and Dex-BSe in the absence or presence of NIR light irradiation was investigated. As shown in Supplementary Fig. 7, limited penetration was observed for Dex nanoparticles with or without NIR irradiation. In contrast, Janus Dex-BSe quickly penetrated into the biofilm after NIR light irradiation was applied for 5 min while no noticeable penetration occurred in the absence of NIR light irradiation. These phenomena indicate that the enhanced penetration mediated by Janus Dex-BSe was attributed to the self-propelled action motion, which is consistent with previous works^{18,54}.

Supplementary Fig. 7 3D CLSM images and corresponding z-stack images of *S. aureus* biofilms treated with Cy5.5-labeled Dex and Dex-BSe at Dex concentration of 307 µg/mL in the presence or absence of NIR light (808 nm, 1.0 W/cm²) irradiation for 1, 3, and 5 min.

To verify the effect of penetration on killing effect, the photothermal killing ability of Dex-BSe against *S. aureus* biofilm after different incubation times was investigated by standard plate counting method. As shown in Supplementary Fig. 18a,b, significant photothermal killing effect of Dex-BSe against *S. aureus* biofilm was observed after 10 min of incubation. As the incubation time was extended to 60 min, the bacterial viability was further reduced, indicating the enhanced antibiofilm efficacy under NIR light irradiation as the penetration of Janus Dex-BSe in *S. aureus* biofilm was increased.

Related information has been added in **Lines 216-224 and 368-375**, as well as **Supplementary Figs. 7 and 18**.

Supplementary Fig. 18. **a** Representative photographs of corresponding *S. aureus* colonies and **(b)** corresponding bacterial viability of *S. aureus* biofilms treated with Dex-BSe (512 $\mu\text{g/mL}$) under NIR light irradiation (808 nm, 1.0 W/cm^2) for 5 min after different incubation times. Data are presented as mean values \pm SD ($n = 4$ independent samples). Statistical significance was calculated by one-way ANOVA using the Tukey post-test. Source data are provided as a Source Data file.

2. Only Gram-positive bacteria S. aureus and MRSA were employed to demonstrate biofilm dispersion by Dex-BS. It would be beneficial to explore the dispersion effect on biofilms of Gram-negative bacteria.

Response: We are grateful to the reviewer for the comment. As suggested, the dispersion effect on biofilms of Gram-negative *E. coli* mediated by Dex-BSe was further explored by crystal violet assay. As shown in Supplementary Fig. 9, the biofilm biomass exhibited negligible changes after treatment with different concentrations of Dex-BSe, indicating negligible dispersion effect on biofilms of Gram-negative bacteria, which is consistent with previous report²⁴. Related information has been added in **Lines 242-247**, as well as **Supplementary Fig 9**.

Supplementary Fig. 9. Quantitative analysis of the crystal violet-stained *E. coli* biofilms treated with Dex-BSe at different concentrations. Data are presented as mean values \pm SD ($n = 4$ independent samples). Source data are provided as a Source Data file.

3. For RNA-seq transcriptomics, it was found that the difference in gene expression identified in the Dex-BS group was more pronounced than in the Dex group. To further understand the underlying mechanism, DEGs between BS and the control group should also be analyzed to provide information whether BS has effect on biofilm-related genes.

Response: We appreciate the comment from the reviewer. As suggested, to investigate the effect of BSe on biofilm-related genes, DEGs between the BSe group and the control group were analyzed. It was found that 123 DEGs were upregulated and 201 DEGs were downregulated in the BSe group compared with the control group (Supplementary Fig. 14a). KEGG enrichment analysis demonstrates that DEGs between the BSe group and control group were related to quorum sensing (Supplementary Fig. 14b), which may contribute to the biofilm dispersion mediated by Dex-BSe. Corresponding information has been added in **Lines 317-323** and **Supplementary Fig. 14**.

Supplementary Fig. 14. (a) Volcano plot and (b) KEGG enrichment analysis of DEGs in biofilms after treatment with PBS (control) and BSe.

4. In Fig 5c, the authors evaluated the photothermal killing effect of Dex-BS using

CLSM. It is also necessary to perform quantitative analysis, such as CFU counts and biofilm biomass.

Response: We appreciate the comments from the reviewer. As suggested, quantitative analysis has been performed to evaluate the photothermal killing effect of Dex-BSe. The quantitative analysis employing crystal violet-staining and standard plate counting method confirmed the substantially higher antibiofilm effect of Dex-BSe than BSe and CS-BSe nanoparticles (Supplementary Fig 17). Corresponding information has been added in **Lines 363-367** and **Supplementary Fig. 17**.

Supplementary Fig. 17. **a** Quantitative analysis of the crystal violet-stained *S. aureus* biofilms and **(b)** bacterial viability of *S. aureus* biofilms treated with PBS (control), BSe, CS-BSe and Dex-BSe, respectively under 808 NIR light (1.0 W/cm², 5 min) irradiation by standard plate counting method. Data are presented as mean values ± SD ($n = 3$ independent samples). Statistical significance was calculated by one-way ANOVA using the Tukey post-test. Source data are provided as a Source Data file.

5. To verify the synergistic effect of biofilm dispersion and photothermal killing mediated by Dex-BS, the combination index or degree of synergy should be calculated.

Response: We are grateful to the reviewer for the comment. As suggested, the degree of synergy (S) was calculated to verify the synergistic effect of biofilm dispersion and photothermal killing mediated by Dex-BSe. Judging from Supplementary Fig. 6c, a synergistic effect of biofilm dispersion between the Dex and BSe components in Janus Dex-BSe nanoparticles was achieved ($S > 0$)⁵⁵. Similarly, since almost no photothermal killing effect of Dex was observed (Fig. R11), $S > 0$ could be calculated, verifying the synergistic effect of photothermal killing mediated by Dex-BSe. Corresponding information has been added in **Lines 264-265**.

Fig. R11. Bacterial viability of *S. aureus* biofilms treated with PBS (control) and Dex respectively under 808 NIR light (1.0 W/cm², 5 min) irradiation by standard plate counting method. Data are presented as mean values \pm SD ($n = 3$ independent samples).

6. Some citation numbers of the references are not superscripted, such as Nos. 19, 22, and 58.

Response: We appreciate the comments from the reviewer. As suggested, all of citation numbers of the references have been carefully checked to improve the quality of the manuscript.

7. It would be better to use “Dex-BSe” as the abbreviation of dextran-bismuth selenide.

Response: We thank the reviewer for the comment. As suggested, “Dex-BSe” has been selected as the abbreviation of dextran-bismuth in the revised manuscript.

8. In Fig. 7a, “removel” should be “removal”.

Response: We are grateful to the reviewer and apologize for the typo. All of the typos have been corrected to improve the quality of the manuscript.

We greatly thank all the reviewers’ valuable comments, which help us substantially improved our manuscript. We hope that the revised manuscript will prove to be acceptable for publication in *Nature Communications*.

Sincerely yours,
Dr. Xu on behalf of all authors

Reviewers' Comments:

Reviewer #1:

Remarks to the Author:

Thanks for the comprehensive answers. I recommend your revised manuscript for publication.

Reviewer #2:

Remarks to the Author:

The previous critiques were all addressed including additional and relevant experiments with thoughtful responses and discussions. A minor point is to include the CFU data in Fig 3d instead of photographs of the plates (which are subjective).

Reviewer #3:

Remarks to the Author:

The authors addressed all the questions raised by the reviewers appropriately, I recommend the manuscript for publication in Nature Communications.

Response to Reviewers

From Reviewer #1 (Remarks to the Author):

Thanks for the comprehensive answers. I recommend your revised manuscript for publication.

Response: We appreciate the positive comment from the reviewer.

From Reviewer #2 (Remarks to the Author):

The previous critiques were all addressed including additional and relevant experiments with thoughtful responses and discussions. A minor point is to include the CFU data in Fig 3d instead of photographs of the plates (which are subjective).

Response: We thank the reviewer for the valuable comments. As suggested, the CFU data have been included in **Fig. 3d**.

Fig. 3d Bacterial counts in *S. aureus* biofilms treated with CS-BSe and Dex-BSe for 2, 4, and 6 h, respectively, with photographs of bacterial cultures shown inset. Data are presented as mean values \pm SD ($n = 3$ independent samples).

From Reviewer #3 (Remarks to the Author):

The authors addressed all the questions raised by the reviewers appropriately, I recommend the manuscript for publication in Nature Communications.

Response: We thank the reviewer for the positive comment.

We greatly thank all the reviewers' valuable comments, which help us substantially improved our manuscript.